# Historical Suitability and Sustainability of Sicani Mountains Landscape (Western Sicily): An Integrated Approach of Phytosociology and Archaeobotany

**Giuseppe Bazan** [1], **Claudia Speciale** [2], **Angelo Castrorao Barba** [3,*], **Salvatore Cambria** [4], **Roberto Miccichè** [5] **and Pasquale Marino** [6]

1   Department of Biological, Chemical and Pharmaceutical Sciences and Technologies (STEBICEF), University of Palermo, 90123 Palermo, Italy; giuseppe.bazan@unipa.it
2   National Institute of Geophysics and Volcanology, 80124 Napoli, Italy; claudia.speciale@ingv.it
3   CSIC (Consejo Superior de Investigaciones Científicas-Spanish National Research Council), EEA Escuela de Estudios Árabes, 18010 Granada, Spain
4   Department of Geological, Biological and Environmental Sciences, University of Catania, 95124 Catania, Italy; cambria_salvatore@yahoo.it
5   Institute for Digital Exploration (IDEx)—History Department, South Florida University, Tampa, FL 33620, USA; robertomicciche@gmail.com
6   Bona Furtuna LLC, Los Gatos, CA 95030, USA; marino@bonafurtuna.com
*   Correspondence: castroraobarba@eea.csic.es

**Abstract:** Since 2015, the ongoing project "Harvesting Memories" has been focused on long-term landscape dynamics in Sicani Mountains (Western Sicily). Archaeological excavations in the case study site of Contrada Castro (Corleone) have investigated a settlement which was mainly occupied during the Early Middle Ages (late 8th–11th century AD). This paper aims to understand the historical suitability and sustainability of this area analysing the correlation between the current dynamics of plant communities and the historical use of woods detected by the archaeobotanical record. An integrated approach between phytosociology and archaeobotany has been applied. The vegetation series of the study area has been used as a model to understand the ecological meaning and spatial distribution of archaeobotanical data on charcoals from the Medieval layers of the Contrada Castro site. The intersection between the frequency data of the archaeobotanical record and the phytosociological analysis have confirmed the maintenance of the same plant communities during the last millennium due to the sustainable exploitation of wood resources. An integrated comparison between the structure and composition of current phytocoenoses with archaeobotanical data allowed us to confirm that this landscape is High Nature Value (HNV) farmland and to interpret the historical vegetation dynamics linked to the activities and economy of a rural community.

**Keywords:** historical ecology; landscape archaeology; vegetation science; anthracology; vegetation series; Mediterranean woods; high nature value (HNV) farmlands; historical landscapes; early middle ages

## 1. Introduction

In long-term anthropized landscapes, the biodiversity has been preserved over time thanks to agricultural, pastoral and silvicultural practices that we would define today as sustainable. The high biodiversity, maintained during the long interaction between natural processes and human history, is a peculiarity of certain types of landscapes and agrarian systems that, for this reason, are recognized as High Nature Value (HNV) farmlands [1]. The strong connection between biodiversity conservation

and the maintenance of traditional agricultural activities has been the concept behind the development of the definition of HNV farmlands in Europe [2]. In particular, seminatural pastures and woods are essential elements of traditional and historical agricultural landscapes for the maintenance of the high degree of naturalness [3]. High presence of seminatural areas, extensive mosaic landscapes and areas hosting species of conservation concern are elements for the definition of an HNV farmland [4]. In most cases, the HNV farmlands are "historical landscapes" and are intended to be landscapes that are long-standing in a certain territory without or with gradual changes [5,6]. However, what is it meant by "long-standing" in a given area? The understanding of a historical landscape is based on the analysis of natural factors and historical dynamics that have determined its characteristics and on the interpretation of human–environment interactions. In fact, the landscape as a biological and cultural system is the result of processes of changes that have determined a stratified and dynamic pattern over time, notably also in the Mediterranean [7,8].

Vegetation studies are useful to describe landscape patterns and transformations. The structure and floristic compositions of plant communities are strictly connected to the environmental set-up of the ecosystems. For this reason, the pool of community species has been recognized as an indicator of environmental conditions and the characterization of the ecosystems [9–11]. Phytocoenoses are not only the result of natural processes but also derive from the anthropic action that changes their characteristics. Secondary formations such as shrubs, pasture meadows, ruderal and weed communities are examples of vegetation derived from sylvo-pastoral and agricultural activities [12] and in many cases are of considerable phytogeographic interest [13–15]. The structure, diversity and floristic composition of these communities have been influenced by the current and historical use of the territory. In particular, fire, deforestation and grazing are the anthropic activities that have influenced the pattern of natural plant communities present in rural landscapes [16–18]. In summary, the characteristics of a plant association—such as the result of interactions between natural and human factors—are fundamental keys to understanding the landscape. However, this is not sufficient; to gain a wider view of the landscape, it is necessary to interpret the mosaic of associations generated by the anthropic disturbance in terms of series of vegetation [12,19–21]. The study of the landscape from the perspective of vegetation series allows us to interpret the territorial mosaic both in terms of spatial patterns (different associations that share the same ecology) and in the temporal dynamics (the phytocoenosis that occurs over time in relation to anthropic activities). In fact, the spatial distribution of a vegetation series is influenced by environmental factors (e.g., climate, lithology, landforms), which change over the long term, often measured with the geological time scale. The anthropic factors do not change the entire spatial distribution of a series but modify, in a shorter time scale, the plant associations (the different stages) that compose it. However, the diachronic analysis of the landscape in terms of vegetation series—as for all ecosystem ecology analyses [22]—is limited only to a time-range of just over a century [12].

The improvement of the temporal depth in the analysis of landscape dynamics is possible through a Historical Ecology approach that integrates historical and archaeological data with ecological information [23] and specifically—in this "vegetation series perspective"—with the dynamic–catenal phytosociology (see Rivas-Martínez [20]). This multidisciplinary perspective makes it possible to study the interaction between human societies and biophysical environments while also considering long-term dynamics [24,25].

The reconstruction of past landscape dynamics and the impact of agricultural, pastoral and forestry practices on vegetation during the long term can be significantly improved by archaeobotanical data [26]. The impact of anthropic activity on vegetation over the centuries can be assessed using pollen [27,28] but also archaeobotanical macroremains [29,30]. In particular, the study of the charcoals found in archaeological excavations provides a fundamental tool for the reconstruction of past landscapes and for interpreting strategies for the exploitation of forest resources by human communities [31–33].

Archaeobotanical studies and analyses of wood charcoals for the reconstruction of paleoenvironments are a practice in most current archaeological projects in the Mediterranean area [34–36], although in Sicily the picture is still rather limited [37–39]. The presence of wood remains, often charred,

in archaeological contexts is filtered by the process of human selection—for different purposes—of the plant resources available in the surrounding environment [40–42].

The Sicani Mountains, in central-western Sicily, have several historical landscapes with a high degree of biodiversity and a diversified territorial mosaic rich of peculiar phytocoenosis [43–45] that have been recognized by the European Environment Agency as HNV farmland (Figure 1) [46].

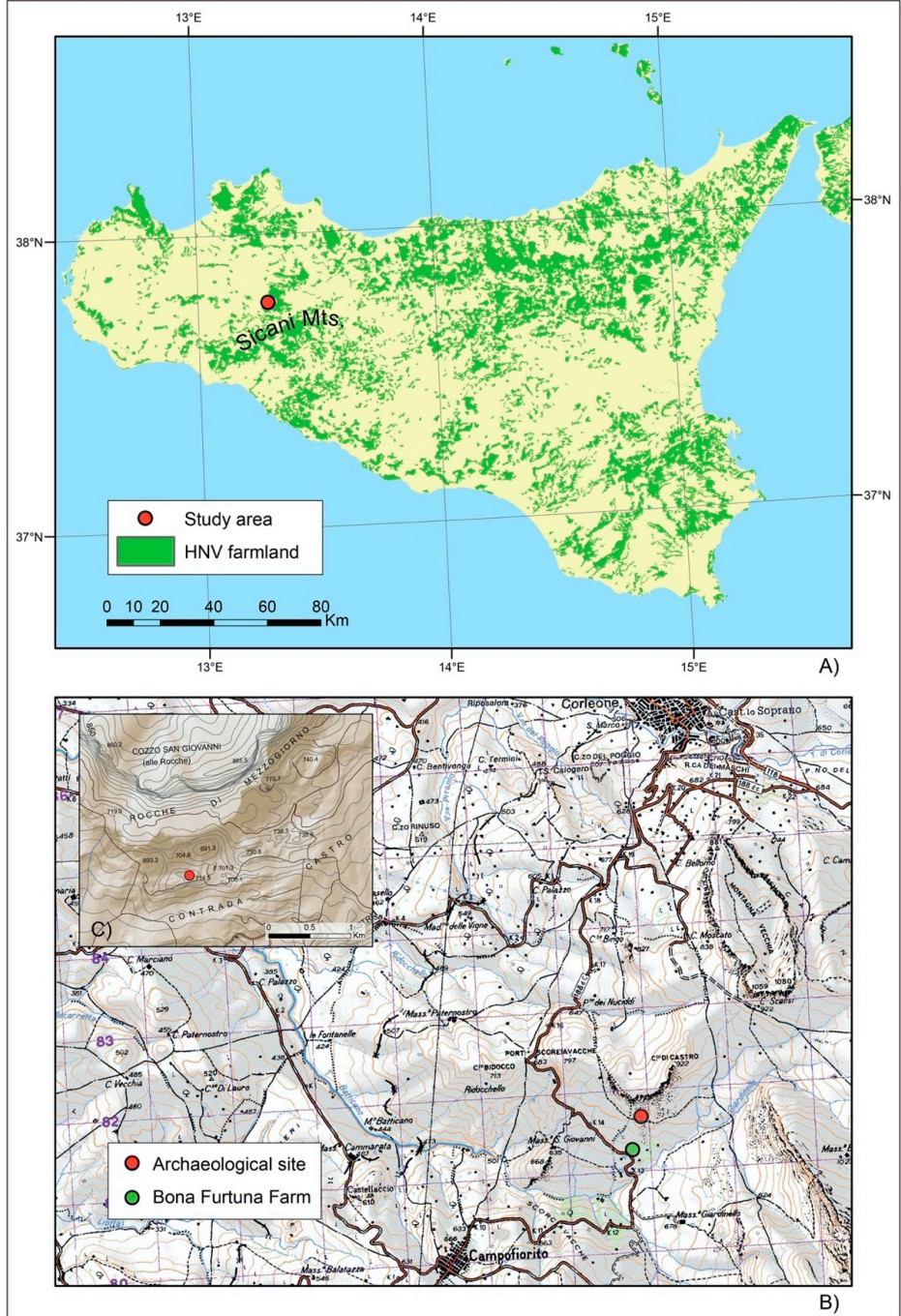

**Figure 1.** (**A**) Case study area and distribution of High Nature Value (HNV) farmland in Sicily (Source: European Environment Agency [46]). (**B**) Location of archaeological site (Contrada Castro) on topographic base map by the Italian Geographic Military Institute (aut. n. 4848 27/07/1998). The U.T.M. grid, in purple, has an interval of 1000 metres. (**C**) Topographical map of the location of the archaeological excavation in Contrada Castro (Redrawn from: Regional Geographical Information System of Sicily—[47]).

Recent research in the Castro-Giardinello area (Corleone, Palermo) [48,49], the study area of this paper, has shown that a small area of about 400 hectares preserves a biodiversity of about 502 taxa of vascular flora that have also been an important resource in traditional food habits [50]. As a result of the dynamics of the renaturalization which has taken place in the last 60 years, this sector of the Sicani Mountains (Figure 2) retains a high degree of naturalness and a low anthropic impact [10]. This interesting biodiversity is linked to a territory with long-term human occupation as evidenced by a recent archaeological survey [48].

The aim of this paper is to detect the relationship between the geobotanical characteristics of the landscape (phytosociological analysis of current vegetation dynamics) and the effective historical use of wood resources (anthropological analysis on charcoal finds from the Medieval phases of Contrada Castro site) in order to understand the long-term suitability and sustainability of this historical landscape and HNV farmland in Sicani Mountains district (central-western Sicily).

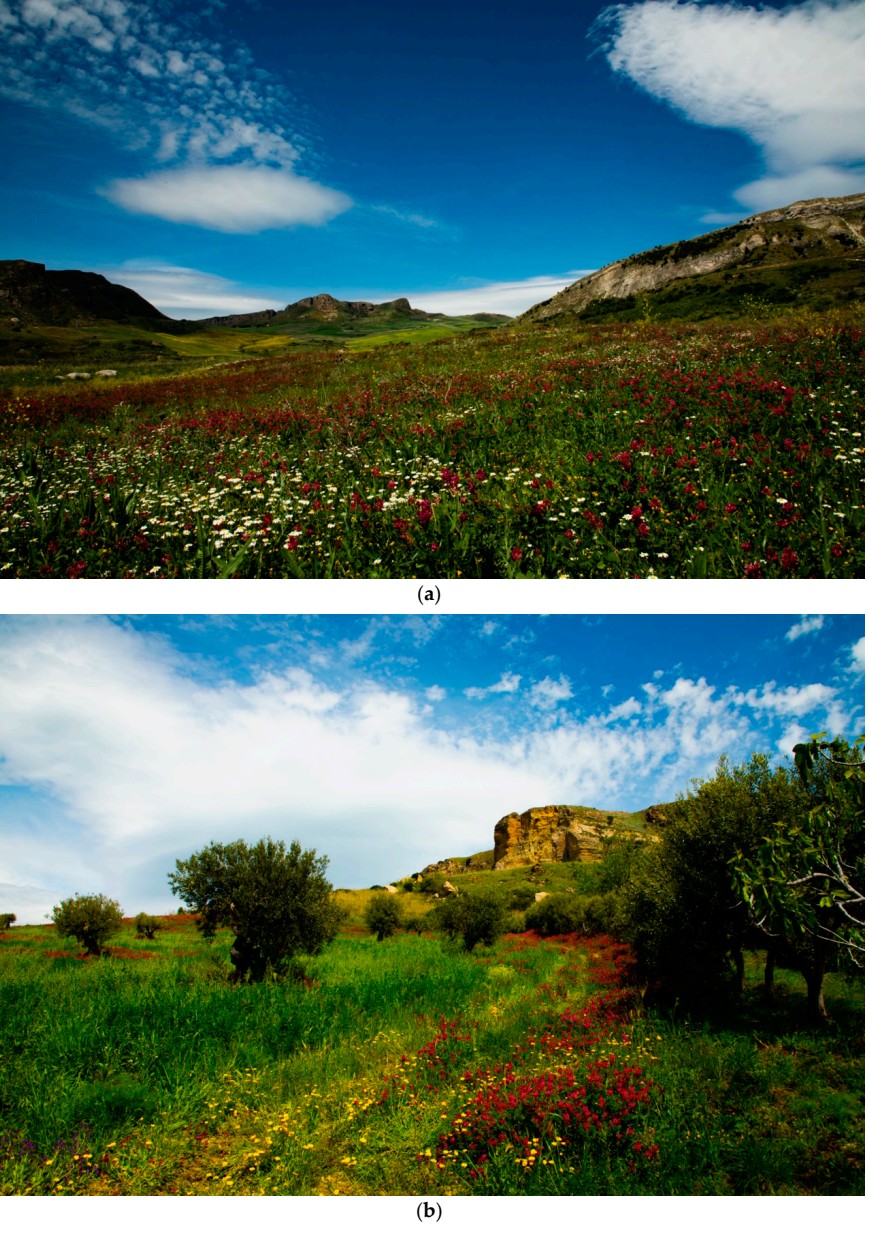

(**a**)

(**b**)

**Figure 2.** Sicani Mountains landscape: Farmland in Contrada Giardinello (**a**) and olive groves (**b**) near Pizzo Castro (Corleone). The area depicted in these photos is free of anthropic elements (buildings, roads, electric lines, etc.). Photos by Pasquale Marino.

## 2. Archaeological Background of the Site

Since 2015, the Castro-Giardinello area located in the Sicani Mountains district, in the southern part of the territory of Corleone (central-western Sicily), has been investigated by archaeological survey and excavation (directed by the Soprintendenza BB.CC.AA. of Palermo) within the project "Harvesting Memories" focused on the investigation of long-term landscape trajectories and human dynamics [49]. In particular, the project area is located in the Bona Furtuna LLC estate (a 100% organic farm) between the valley of the Giardinello creek and the south-western side of Barraù mount (maximum height 1475 m a.s.l.) (Figures 1b and 2). The long-term anthropic occupation of this area has been detected by a field survey that identified various sites dating back to the Bronze Age, classical, medieval and postmedieval periods [48].

For its topographical position on a hill-top plateau (715–713 m a.s.l.) and its remarkable presence of pottery, the site of Contrada Castro has been chosen for archaeological excavations from 2017, and these are still ongoing (Figure 3). The site extends over a flat, raised, east-west plateau (0.54 ha) on brown marls substrate. To the north, it is adjacent to a sinkhole that separates it from the steep slopes of Pizzo Castro, and to the south, there is an almost vertical slope towards the valley of the Giardinello river (Figure 1b).

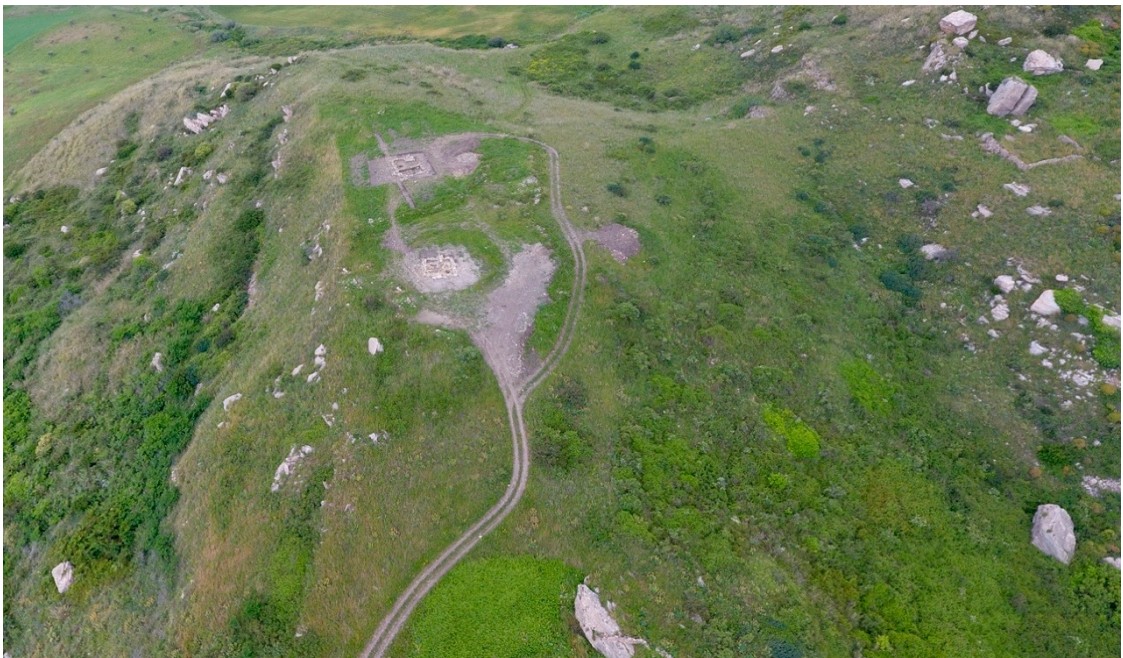

**Figure 3.** The hill-top plateau of the archaeological site of Contrada Castro (Corleone, Palermo, Sicily). Drone photo by Filippo Pisciotta.

In this area, the surveys have highlighted the presence of dispersed ceramic sherds which indicated a potential occupation during the Early Middle Ages (Byzantine and Islamic periods, 7th–11th centuries AD), while rare fragments of black-gloss pottery also indicated an earlier occupation during an archaic/classical period [48]. The initial hypothesis was therefore related to the possible presence of a site that was intensely populated during the Early Middle Ages and subsequently abandoned and never again permanently resettled. The first archaeological seasons have confirmed this hypothesis [48]. The earlier period of the site is documented by some wall remains associated with ceramic finds of the late 6th–5th centuries BC. On the collapses of these structures, some evidence of a reoccupation of the site are related to two perinatal burials dated by radiocarbon analysis between the 7th and the mid-8th centuries AD (sigma 1 65%: AD 662–AD 778). Later, during the late 8th–9th centuries AD, this area was further exploited for the installation of a building dedicated in the first phase to the production of ceramics and tiles, as evidenced by the discovery of two circular kilns. The chronology of these

structure is also based on radiocarbon analysis of *Pistacia terebinthus* wood charcoals (sigma 1 65%: AD 774–AD 878) from a burnt layer related to the productive activity of one of the two kilns. This structure was reused after a short time, changing its function because one of the pottery kilns was converted into a food oven (possibly for bread). The collapse of this building occurred during the 9th century AD. On the ruins of this building, which is no longer visible, another building was built with two phases, which can be placed chronologically between the first half of the 10th and the 11th century AD. For this period, the chronology was confirmed by pottery chronotypology and by radiocarbon analysis of an *Equus asinus* bone (sigma 1 65%: AD 965–AD 1042) [48]. The largest number of archaeobotanical findings comes from the medieval layers of the site and can be subdivided into two periods: Period 1, late 8th–9th centuries AD; Period 2, 10th–11th centuries AD.

## 3. Materials and Methods

### 3.1. Vegetation Data

The syn-phytosociological study of the vegetation allowed us to define a forest landscape model (current and potential) of the Sicani Mountains, which was necessary for the interpretation of anthracological data from the Contrada Castro sequence. The analysis of landscape patterns in terms of vegetation series followed the approach outlined in detail by Bazan et al. [21].

The vegetation surveys were conducted based on the classical phytosociological approach [51] including recent methodological advances [19]. A total of 24 relevés, in 100 m$^2$ plots, were created during the spring 2019 (Table S1). The survey was carried out with a Stratified Random Sampling by dividing the study area into different physiognomical homogeneous contexts, based on the stages of the vegetation series defined by Bazan et al. [12] and randomly sampling within each of these smaller areas [52].

In order to get a more complete picture of the floristic composition and structure of the forest and preforest associations, the relevés of Gianguzzi et al. [53] were also considered. The complete data set consisted of 106 relevés which were summarized in a synoptic table (Table 1).

The mean coverage of each woody species (phanerophytes and nanophanerophytes) within each association was calculated by transforming the Braun-Blanquet cover-abundance scale into a quantitative form (0–100%) as proposed by McNellie et al. [54]. A mean coverage curve was constructed for the associations of the vegetation series describing our forest landscape model.

The syntaxonomical nomenclature followed the "International Code of Phytosociological Nomenclature" [55]. Species identification was done using the identification key of Pignatti et al. [56] and for the nomenclature, the online database "The Plant List" [57], was used. The complete names of specific and subspecific taxa of this paper, including the authors' citations of plant names, are given in Table 1.

The bioclimatic data followed the framework developed for Sicily by Bazan et al. [58]; other ecological information was obtained from Gianguzzi et al. [59].

**Table 1.** Synoptic table of the associations of vegetation series. Column numbers: 1 = *Ampelodesmo mauritanici-Quercetum ilicis*; 2 = *Sorbo torminalis-Quercetum ilicis*; 3 = *Oleo oleaster-Quercetum virgilianae*; 4 = *Sorbo torminalis-Quercetum virgilianae*; 5 = *Euphorbio characiae-Prunetum spinosae*; 6 = *Roso siculae-Prunetum spinosae*; 7 = *Roso corymbiferae-Rubetum ulmifolii*; 8 = *Crataegetum laciniatae*; 9 = *Ulmo-Salicetum pedicellatae*. Roman numbers indicate the presence classes of species in the phytosociological relevés of Table S1 as follows: I = 0–20%, II = 20–40%, III = 40–60%, IV = 60–80%, V = 80–100%. The values of diagnostic species of association and alliance are in the grey boxes.

| Column Number | 1 | 2 | 3 | 4 | 5 | 6 | 7 | 8 | 9 |
|---|---|---|---|---|---|---|---|---|---|
| **Total Number of Relevés** | 27 | 18 | 5 | 9 | 14 | 8 | 8 | 7 | 10 |
| **Characteristics and diff. of ass. and subass.** ***Ampelodesmo mauritanici-Quercetum ilicis*** | | | | | | | | | |
| *Quercus ilex* L. | V | V | II | IV | I | . | . | . | . |
| *Hippocrepis emerus* (L.) Lassen subsp. *emeroides* (Boiss. & Spruner) Lassen | IV | . | IV | . | I | . | . | . | II |
| *Ampelodesmos mauritanicus* (Poir.) T. Durand & Schinz. | IV | II | II | II | III | . | III | . | . |
| *Lonicera implexa* Aiton | IV | II | . | . | . | . | . | . | . |
| *Viburnum tinus* L. | III | . | . | I | . | . | . | . | . |
| *Arbutus unedo* L. | III | I | . | . | . | . | . | . | . |
| **Characteristics and differentials of ass. and subass.** ***Sorbo torminalis-Quercetum ilicis*** | | | | | | | | | |
| *Acer campestre* L. | . | V | . | II | . | I | . | III | . |
| *Euphorbia amygdaloides* subsp. *arbuscula* Meusel | I | III | . | V | I | II | V | II | . |
| *Brachypodium sylvaticum* (Huds.) P. Beauv. | IV | III | . | V | III | V | V | IV | . |
| *Daphne laureola* L. | I | II | . | IV | I | V | III | IV | . |
| **Characteristics and differentials of ass. and subass.** ***Oleo oleaster-Quercetum virgilianae*** | | | | | | | | | |
| *Quercus virgiliana* (Ten.) Ten. | IV | II | V | V | . | . | . | . | . |
| *Prasium majus* L. | II | I | II | . | . | . | . | . | . |
| **Characteristics and differentials of ass.** ***Sorbo torminalis-Quercetum virgilianae*** | | | | | | | | | |
| *Sorbus torminalis* (L.) Crantz | . | III | . | V | . | . | . | . | . |
| **Characteristics of the alliance** ***Fraxino-Quercion ilicis*** **and of the upper units** | | | | | | | | | |
| *Asparagus acutifolius* L. | V | IV | III | IV | IV | III | V | II | . |
| *Dioscorea communis* (L.) Caddick & Wilkin | V | IV | . | IV | II | . | V | I | V |
| *Ruscus aculeatus* L. | V | IV | V | IV | IV | . | V | II | II |
| *Smilax aspera* L. | V | III | IV | II | II | . | IV | I | . |
| *Rubia peregrina* L. subsp. *longifolia* (Poir.) O. Bolòs | V | III | IV | IV | IV | . | IV | . | IV |
| *Cyclamen hederifolium* Aiton | V | II | III | IV | . | . | II | I | . |
| *Fraxinus ornus* L. | IV | IV | I | II | . | I | . | . | . |
| *Allium subhirsutum* L. | IV | III | . | IV | III | . | . | II | . |
| *Festuca drymeja* Mert. & W.D.J.Koch. | III | II | . | IV | . | . | . | . | . |
| *Osyris alba* L. | III | I | III | . | III | . | . | . | V |
| *Pistacia terebinthus* L. | III | . | . | . | . | . | . | . | . |
| *Cyclamen repandum* Sm. | II | IV | V | III | . | . | . | . | . |
| *Rosa sempervirens* L. | II | I | I | III | II | . | I | . | . |
| *Teucrium flavum* L. | II | I | I | . | . | . | . | . | . |
| *Pistacia lentiscus* L. | II | . | . | . | . | . | . | . | . |
| *Lonicera etrusca* Santi | I | V | II | IV | II | I | IV | II | . |
| *Cistus creticus* L. subsp. *creticus* | I | II | III | II | . | . | . | . | . |
| *Viola alba* Besser. subsp. *dehnhardtii* (Ten.) W. Becker | I | I | I | . | . | . | . | II | . |
| *Arisarum vulgare* Targ.Tozz. | I | I | . | II | III | . | . | . | . |
| *Carex distachya* Desf. | I | I | . | II | . | . | . | . | . |
| *Clinopodium vulgare* L. subsp. *orientale* Bothmer | I | . | II | . | . | . | II | . | . |
| *Pulicaria odora* (L.) Rchb. | I | . | . | I | . | . | . | . | . |

**Table 1.** *Cont.*

| Column Number | 1 | 2 | 3 | 4 | 5 | 6 | 7 | 8 | 9 |
|---|---|---|---|---|---|---|---|---|---|
| **Total Number of Relevés** | 27 | 18 | 5 | 9 | 14 | 8 | 8 | 7 | 10 |

**Characteristics of the alliance *Fraxino-Quercion ilicis* and of the upper units**

| | | | | | | | | | |
|---|---|---|---|---|---|---|---|---|---|
| *Chamaerops humilis* L. | I | . | . | . | . | . | . | . | . |
| *Cephalanthera longifolia* (L.) Fitsch | I | . | . | . | . | . | . | . | . |
| *Daphne gnidium* L. | I | . | . | . | . | . | . | . | . |
| *Polystichum setiferum* (Forssk.) Moore ex Woyn | I | . | . | . | . | . | . | . | . |
| *Myrtus communis* L. | I | . | . | . | . | . | . | . | . |
| *Anthriscus nemorosa* (M. Bieb.) Spreng. | . | II | . | II | . | . | II | I | . |
| *Asplenium onopteris* L. | . | I | II | . | . | . | . | . | . |
| *Calicotome infesta* (C. Presl) Guss. subsp. *infesta* | . | . | I | . | . | . | . | . | . |
| *Quercus amplifolia* Guss. | . | . | . | I | . | . | . | . | . |

**Char. and diff. of ass.*Euphorbio characiae-Prunetum spinosae*and *Roso siculae-Prunetum spinosae***

| | | | | | | | | | |
|---|---|---|---|---|---|---|---|---|---|
| *Euphorbia characias* L. | III | II | . | . | V | . | . | . | . |
| *Prunus spinosa* L. | I | II | I | II | IV | V | V | III | . |
| *Rosa sicula* Tratt. | . | . | . | II | . | V | . | IV | . |
| *Rosa glutinosa* Sibth. & Sm. | . | . | . | . | . | V | . | II | . |

**Characteristics and differentials of ass. *Roso corymbiferae–Rubetum ulmifolii***

| | | | | | | | | | |
|---|---|---|---|---|---|---|---|---|---|
| *Rubus ulmifolius* Schott | II | II | IV | IV | V | IV | V | I | II |
| *Crataegus monogyna* Jacq. var. *monogyna* | IV | III | I | III | IV | . | V | III | . |
| *Paeonia mascula* (L.) Mill. subsp. *russoi* (Biv.) Cullen & Heywood | II | II | . | IV | . | II | IV | III | . |
| *Rosa corymbifera* Borkh. | . | I | . | . | III | . | V | III | . |

**Characteristics and differentials of ass. *Crataegetum laciniatae***

| | | | | | | | | | |
|---|---|---|---|---|---|---|---|---|---|
| *Crataegus rhipidophylla* Gand. | . | . | . | . | . | V | II | V | . |

**Characteristics and differentials of the alliance *Pruno–Rubion ulmifolii* and of the upper units**

| | | | | | | | | | |
|---|---|---|---|---|---|---|---|---|---|
| *Hedera helix* L. subsp. *helix* | V | V | II | IV | IV | II | V | III | II |
| *Rosa canina* L. | II | II | III | III | V | V | V | II | . |
| *Pyrus spinosa* Forssk. | . | . | . | . | III | I | V | IV | . |
| *Rosa micrantha* Sm. ex Sm. | . | . | . | . | . | . | II | . | . |
| *Rosa balsamica* Besser | . | . | . | . | . | . | I | . | . |
| *Clematis vitalba* L. | III | II | I | I | I | III | . | I | . |
| *Rhamnus alaternus* L. | . | . | . | . | I | . | . | . | . |

**Characteristics and differentials of the alliance *Berberido aetnensis–Crataegion laciniatae***

| | | | | | | | | | |
|---|---|---|---|---|---|---|---|---|---|
| *Lamium flexuosum* Ten. | . | . | . | . | . | II | . | II | . |
| *Rubus canescens* DC. | . | I | . | III | . | III | . | III | . |

**Char. and diff. of *Ulmo-Salicetum pedicellatae* and the alliance *Populion albae* and upper**

| | | | | | | | | | |
|---|---|---|---|---|---|---|---|---|---|
| *Salix pedicellata* Desf. | . | . | . | . | . | . | . | . | V |
| *Ulmus minor* Mill. | . | I | . | . | I | . | . | . | IV |
| *Populus nigra* L. | . | . | . | . | . | . | . | . | V |
| *Hypericum hircinum* L. subsp. *majus* (Aiton) N. Robson | . | . | . | . | . | . | . | . | II |
| *Solanum dulcamara* L. | . | . | . | . | . | . | . | . | II |

**Other species**

| | | | | | | | | | |
|---|---|---|---|---|---|---|---|---|---|
| *Sorbus domestica* L. | . | II | I | . | . | . | . | . | . |
| *Erica multiflora* L. subsp. *multiflora* | II | I | . | . | . | . | . | . | . |
| *Buglossoides purpurocaerulea* (L.) I.M.Johnst. | II | I | II | . | II | . | . | . | . |
| *Anthyllis vulneraria* L. subsp. *busambarensis* (Lojac.) Pignatti | I | . | . | . | . | . | . | . | . |

**Table 1.** *Cont.*

| Column Number | 1 | 2 | 3 | 4 | 5 | 6 | 7 | 8 | 9 |
|---|---|---|---|---|---|---|---|---|---|
| **Total Number of Relevés** | **27** | **18** | **5** | **9** | **14** | **8** | **8** | **7** | **10** |
| **Other species** | | | | | | | | | |
| *Achillea ligustica* All. | . | . | . | II | . | I | IV | III | . |
| *Opopanax chironium* (L.) W.D.J. Koch | . | I | . | . | I | III | . | I | . |
| *Malus sylvestris* Mill. | . | . | . | II | . | I | . | . | . |
| *Euonymus europaeus* L. | . | . | . | II | . | . | . | . | . |
| *Cytisus villosus* Pourr. | . | . | . | I | . | . | . | . | . |
| *Mespilus germanica* L. | . | . | . | I | . | . | . | . | . |
| *Sorbus graeca* (Lodd. ex Spach) Kotschy | . | . | . | II | . | . | . | . | . |
| *Acanthus mollis* L. | III | . | III | . | . | . | . | . | . |
| *Arum italicum* Mill. subsp. *italicum* | II | II | . | . | II | . | II | . | IV |
| *Geranium lucidum* L. | . | III | . | . | . | . | . | II | . |
| *Geranium robertianum* L. | I | . | . | II | . | . | . | . | . |
| *Magydaris pastinacea* (Lam.) Paol. | I | . | . | . | II | . | . | . | . |
| *Reichardia picroides* (L.) Roth | I | . | . | . | . | . | . | . | . |
| *Ranunculus pratensis* C.Presl | I | . | . | . | . | . | . | . | . |
| *Thalictrum calabricum* Spreng. | . | II | II | II | . | II | . | I | . |
| *Geum urbanum* L. | . | I | . | . | . | . | . | . | . |
| *Galium aparine* L. | . | I | . | . | I | I | . | II | . |
| *Centranthus ruber* (L.) Dc. subsp. Ruber | I | . | . | . | . | . | . | . | . |
| *Picris hieracioides* Sibth. & Sm. | I | . | . | . | . | . | III | . | . |
| *Asperula laevigata* L. | . | I | . | . | . | . | . | . | . |
| *Conium maculatum* L. | . | I | . | . | . | . | . | I | . |
| *Erysimum bonannianum* C.Presl | . | I | . | . | . | . | . | . | . |
| *Bituminaria bituminosa* (L.) C.H.Stirt. | . | . | II | . | . | . | . | . | . |
| *Rhus coriaria* L. | . | . | I | . | . | . | . | . | II |
| *Hypericum perforatum* L. | . | . | I | . | I | . | . | . | . |
| *Ranunculus bulbosus* L. | . | . | I | . | . | . | . | . | . |
| *Leontodon siculus* (Guss.) Nyman | . | . | I | . | I | III | . | II | . |
| *Agrimonia eupatoria* L. | . | . | I | . | . | . | . | . | . |
| *Anagyris foetida* L. | . | . | . | . | I | . | . | . | . |
| *Arrhenatherum elatius* (L.) J. & C. Presl subsp. *bulbosum* (Willd.) Schübler & G. Martens | . | . | . | . | . | II | . | . | . |
| *Asphodeline lutea* (L.) Rchb. | . | . | . | . | II | . | II | . | . |
| *Clinopodium nepeta* (L.) Kuntze. | . | . | . | . | . | II | . | . | . |
| *Cerastium glomeratum* Thuiil. | . | . | . | . | . | . | . | II | . |
| *Crataegus laevigata* Jacq. | . | . | . | . | I | . | . | . | . |
| *Dactylis glomerata* L. subsp. *hispanica* (Roth) Nyman | . | . | . | . | I | . | . | . | . |
| *Elymus panormitanus* (Parl.) Tzvelev | . | . | . | . | . | I | . | II | . |
| *Euphorbia ceratocarpa* Ten. | . | . | . | . | III | . | I | . | II |
| *Fedia graciliflora* Fisch. & C.A. Mey. | . | . | . | . | . | . | . | II | . |
| *Ficaria verna* Huds. | . | . | . | . | . | . | . | II | . |
| *Hyoseris radiata* L. | . | . | . | . | II | . | . | . | . |
| *Ostrya carpinifolia* Scop. | . | . | . | II | . | . | . | . | . |
| *Piptatherum miliaceum* (L.) Coss. | . | . | . | . | I | . | I | . | . |
| *Pteridium aquilinum* (L.) Kuhn subsp. *aquilinum* | . | . | . | . | . | . | I | . | . |
| *Rumex thyrsoides* Desf. | . | . | . | . | I | . | . | . | . |
| *Salvia argentea* L. | . | . | . | . | . | I | . | . | . |
| *Sanguisorba minor* Scop. | . | . | . | . | . | I | . | . | . |
| *Scandix pecten-veneris* L. | . | . | . | . | . | . | . | I | . |
| *Festuca arundinacea* Schreb. | . | . | . | . | I | . | . | . | . |
| *Senecio squalidus* L. subsp. *microglossus* (Guss.) Arcang. | . | . | . | . | . | I | . | . | . |

**Table 1.** *Cont.*

| Column Number | 1 | 2 | 3 | 4 | 5 | 6 | 7 | 8 | 9 |
|---|---|---|---|---|---|---|---|---|---|
| **Total Number of Relevés** | 27 | 18 | 5 | 9 | 14 | 8 | 8 | 7 | 10 |
| **Other species** | | | | | | | | | |
| *Silene vulgaris* (Moench) Garcke | . | . | . | . | . | I | . | . | . |
| *Smyrnium perfoliatum* L. | . | . | . | . | . | . | . | II | . |
| *Smyrnium rotundifolium* Mill. | . | . | . | . | . | . | II | . | . |
| *Vicia villosa* Roth subsp. *villosa* | . | . | . | . | II | . | . | . | . |
| *Vicia villosa* subsp. *varia* (Host) Corb. | . | . | . | . | . | . | II | . | . |
| *Carex pendula* Huds. | . | . | . | . | . | . | . | . | II |
| *Equisetum telmateia* Ehrh. | . | . | . | . | . | . | . | . | V |
| *Eupatorium cannabinum* L. | . | . | . | . | . | . | . | . | IV |
| *Allium triquetrum* L. | . | . | . | . | . | . | . | . | II |
| *Viola alba* Besser subsp. *alba* | . | . | . | . | . | . | . | . | II |

### 3.2. Archaeobotany

The soil samples in this investigation were selectively collected during the 2017 and 2018 excavation campaigns.

For each archaeological layer—also defined by Harris [60] units of stratification (US)—, soil samples equal to about five litres were taken manually and randomly, while only in a particular case—that is, the layer of burnt soil relating to the early medieval furnace—was a total sampling of the layer carried out. The stratigraphic units were analyzed, dividing them by the three chronological phases, but this analysis subsequently focused only on the two medieval phases.

The samples were water floated at the Botanical Garden of Palermo, using two different sieves (5 and 1 mm) to separate the clayey matrix of the soil. Subsequently, the samples were screened using a binocular microscope at the Department of Biological Chemical and Pharmaceutical Sciences and Technologies (STEBICEF) of the University of Palermo.

For each stratigraphic unit, quantitative data (absolute number of fragments) and volumetric data (volume of each wood charcoal remain) for each species were measured. The comparison between the quantitative and volumetric numerical data resulted in a deviation of up to 16% per species. In most cases, for small quantities, this could be considered negligible, while in large quantities, the difference increased exponentially; for example, in the case of *Quercus ilex* in the second phase.

Only a significant fraction of the stratigraphic unit was observed by microscopic analysis rather than the whole record, considering that the variability in excess of 13–15 wood charcoals usually did not increase [41].

A total of 436 wood charcoals were observed for the three archaeological phases, and among them, 346 samples were related to the two medieval phases analyzed in this work. For the Period 1 (late 8th–9th centuries AD), 276 samples were analyzed (from earlier to later: Phase 1, US 43, 47, 49, 52; Phase 2, US 14, 40, 42, 50, 61; Phase 3, US 9, 12, 37; Phase 4, US 8, 36) and 70 samples (US 1, 2) for the Period 2 (10th–11th centuries AD). The chronology of these phases was based on stratigraphic sequence and radiocarbon analysis. Of these, 19% were not taxonomically identified (due to the small size of the fragments in some US or the state of preservation, where combustion had strongly deformed the xylem).

To identify the genera/species, a comparison was made with the reference atlases [61,62], with some scientific papers and with the two available online tools—InsideWood [63] and Microscopic Wood Anatomy [64]—as well as with some reference samples which were not available for all species.

Each sample was observed in the three transversal, tangential and radial sections at a maximum of 40x and measured to obtain a value for the total volumes by species, and therefore their incidence not only numerically but also regarding their actual size with reference to the total was measured. Whenever possible, the diameters of some small branches were measured.

## 4. Results

### 4.1. The Current Plant Landscape

The natural vegetation of this area is composed of remnant forest patches, shrublands and extensive grasslands which are stages of degradation of pre-existing forest due to anthropic activities. Five different woods have been identified according to the environmental characteristics (bioclimate belts and substrate). The main forests formation are holm oak woods (*Ampelodesmo mauritanici-Quercetum ilicis* and *Sorbo torminalis-Quercetum ilicis*) and downy oak woods (*Oleo oleaster-Quercetum virgilianae* and *Sorbo torminalis-Quercetum virgilianae*) (Figure 4). Along the rivers also grow hygrophilous forests with poplars and willows (*Ulmo canescentis-Salicetum pedicellatae*). Shrublands of thorny bushes of Rosaceae and other evergreen Mediterranean bushes are the result of wood degradation due to cutting or fire. Grazing and fire normally lead to the establishment of *Ampelodesmos mauritanicus* grassland. These natural and seminatural vegetation patches are embedded in a matrix of agricultural land uses characterized by cereals, grapevines and olive trees, "the Mediterranean triad" [65].

The syn-phytosociological analysis allowed us to characterize the associations from a floristic and structural point of view (Table 1) and to define the dynamic relationships among seral stages (Figure 4). Four climatophilous vegetation series and one edapho-hygrophilous vegetation series, which together define the forest landscape of this area of the Sicani Mountains, were identified.

For each of them, the association head of series, the main seral stages and the ecological and geographical distribution in the region are described below.

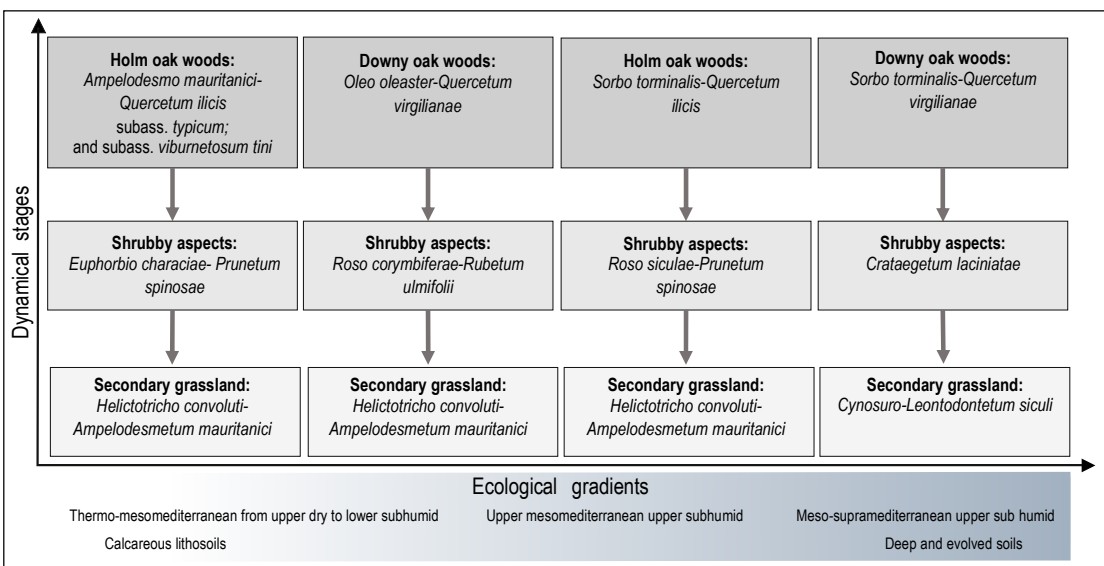

**Figure 4.** Dynamical and catenal contacts among forest associations of the Sicani Mountains.

### 4.1.1. Mesomediterranean Series of Holm Oak (*Ampelodesmo mauritanici-Querco ilicis* Sigmetum)

*Ampelodesmo mauritanici-Quercetum ilicis* Gianguzzi, Cuttonaro, Cusimano & Romano 2016 is a woody vegetation whose tree layer is clearly dominated by *Quercus ilex* and sometimes associated with *Quercus virgiliana* and *Fraxinus ornus* (Column 1 in Table 1). From the floristic point of view, this community is characterized by the occurrence of *Ampelodesmos mauritanicus*, *Hippocrepis emerus* subsp. *emeroides* and *Lonicera implexa*. In addition to the typical aspect, the subass. *viburnetosum tini* Gianguzzi et al. 2016 was surveyed in the more humid and cool stands.

This association represents the most evolved stage of an edapho-climatophilous series that includes various secondary communities derived from the degradation of woodlands. In particular, on quite sloped and sunny stands, it is often replaced by a low shrubby vegetation characterized by

*Euphorbia characias* and *Anagyris foetida*, referable to *Euphorbio characiae-Anagyridetum phoetidis* Gianguzzi, Cuttonaro, Cusimano & Romano 2016, while a further community, named *Euphorbio characiae-Prunetum spinosae* Gianguzzi, Cuttonaro, Cusimano & Romano 2016, is linked to very eroded and less inclined surfaces (Column 5 in Table 1). The degradation of these communities favors the settlement of the dry grasslands belonging to *Helictotricho convolute-Ampelodesmetum mauritanici* Minissale 1994.

This vegetation occurs in the mesomediterranean belt with an upper subhumid ombrotype, generally at an altitude from 450 to 1000 m. It prefers the most inclined slopes with northern exposure, colonizing calcareous lithosols with a significant detrital component derived from erosion and landslides. Currently, this series is known only from the Sicani mountains [52].

### 4.1.2. Supramediterranean Series of Holm Oak (*Sorbo torminalis-Querco ilicis* Sigmetum)

The head series is represented by mesophilous holm oak woods with a closed structure, referable to the *Sorbo torminalis-Quercetum ilicis* Gianguzzi, Cuttonaro, Cusimano & Romano 2016. From the floristic point of view, the association is characterized by the occurrence of some nemoral species with mesic requirements, such as *Daphne laureola*, *Paeonia mascula* subsp. *russoi* and *Euphorbia amygdaloides* subsp. *arbuscula*, etc. (Column 2 in Table 1).

In addition to *Sorbo torminalis-Quercetum ilicis*, this series is constituted by some secondary stages, which are often very frequent due to the strong anthropization of the investigated area; this vegetation should be represented only in the higher altitudes of the Barraù mountain. *Roso siculae-Prunetum spinosae* Gianguzzi, Cuttonaro, Cusimano & Romano 2016 (Column 6 in Table 1) is an orophilous shrubby community that replaces the woods on very shallow soils, forming a very dense and closed vegetation with *Rosa sicula* and some species belonging to *Berberido aetnensis–Crataegion laciniatae* Gianguzzi et al. 2011, such as *Crataegus rhipidophylla* and *Rubus canescens*. A further degradation of the woody vegetation leads to the establishment of dry grasslands belonging to *Helictotricho convolute-Ampelodesmetum mauritanici* Minissale 1994.

This series is linked to stands with eroded calcareous soils, above 1000 m a.s.l., which are characterized by a meso or supramediterranean bioclimate with an upper subhumid ombrotype. It was surveyed only in the Sicani mountains, representing a geographical vicariant of *Acer campestri-Querco ilicis* sigmetum from the Madonie area. In the investigated area, only small, floristically impoverished relics of this vegetation remain.

### 4.1.3. Thermo-Mesomediterranean Series of Downy Oak (*Oleo oleaster-Querco virgilianae* Sigmetum)

The *Oleo oleaster-Quercetum virgilianae* Brullo 1984 represents the most evolved stage of this series, as a woody vegetation dominated by *Quercus virgiliana* (Ten.) Ten. and *Quercus amplifolia* Guss. (Column 3 in Table 1). It is a deciduous woodland with a rich xeric-thermophilous component belonging to *Quercetalia calliprini* Zohary 1955, represented by *Olea europaea* L. var. *sylvestris* Brot., *Pistacia lentiscus*, *Teucrium fruticans* L., *Prasium majus*, etc. Moreover, *Quercetalia ilicis* Br.-Bl. ex Molinier 1934 is well represented by several species, such as *Quercus ilex*, *Rubia peregrina* subsp. *longifolia*, *Carex distachya* Desf., *Osyris alba*, *Asparagus acutifolius*, *Smilax aspera*, *Calicotome infesta*, *Arisarum vulgare*, *Lonicera implexa*, *Ruscus aculeatus*, etc.

The secondary stages of *Oleo sylvestris-Quercetum virgilianae* include the garrigues of the Cisto eriocephali–Ericion multiflorae Biondi 1997 and the deciduous shrubs of Crataego-Prunetea Tx. 1962, such as *Roso corymbiferae-Rubetum-Rubetum ulmifolii* Gianguzzi, Cuttonaro, Cusimano & Romano 2016 in the investigated area (Column 7 in Table 1). The degradation of these shrubby communities, mainly due to fires, leads to the establishment of dry grasslands belonging to *Avenulo cincinnatae-Ampelodesmion mauritanici* Minissale 1995. The further degradation of the soil due to erosive phenomena determines the establishment of ephemeral communities of *Trachynion distachyae* Rivas-Martinez 1978.

This series occurs on more or less deep and evolved soils, developing on various kinds of substrates (limestone, dolomite, marl, clay, basalt, sandstones, schist, etc.). Generally, its potential area coincides with the places which are most suitable for agricultural activities; for this reason, it is

currently fairly localized and covers only small surfaces. From the bioclimatic perspective, it falls within the thermo-mediterranean belt, with some penetrations in the subhumid mesomediterranean belt. Its distribution range includes Sicily and southern Italy [66].

### 4.1.4. Supramediterranean Series of Downy Oak (*Sorbo torminalis-Querco virgilianae* Sigmetum)

*Sorbo torminalis–Quercetum virgilianae* Gianguzzi, Cuttonaro, Cusimano & Romano 2016 is a woodland dominated by *Quercus virgiliana*, whose mesic character is emphasized by the occurrence of *Sorbus torminalis* and a rich herbaceous component characterized by some rare umbellifers, among them *Physospermum verticillatum* (Waldst. & Kit.) Vis., *Geocaryum cynapioides* (Guss.) Engstrand. and *Cnidium silaifolium* (Jacq.) Simonk. (Column 4 in Table 1).

This association has completely disappeared from the area in which it could potentially be present, such as the summit plains of Monte Barraù. It has been replaced by the *Crataegetum laciniatae* Brullo & Marcenò 1984, an orophilous scrub dominated by *Crataegus rhipidophylla*, which is sometimes associated with *Pyrus spinosa* and scattered shrubby specimens of *Acer campestre* (Column 8 in Table 1). However, larger surfaces are covered by a mosaic of mesophilous meadows belonging to *Cynosuro-Leontodontetum siculi* Brullo & Grillo 1978 and *Cachyretum ferulaceae* Raimondo 1980, while the most eroded surfaces and the rocky outcrops are characterized by the prostrate chamaephytic vegetation of *Carduncello pinnati-Thymetum spinulosi* Brullo & Marcenò in Brullo 1984.

It colonizes deep calcareous soils, preferring the fresh and shady northern slopes at altitudes between 800 and 1400 m a.s.l. From the bioclimatic viewpoint, it is localized from the mesomediterranean to the supramediterranean belt with an upper subhumid ombrotype. This series is restricted to the higher belt of the Sicani area [52].

### 4.1.5. Thermo-Mesomediterranean Edapho-Hygrophilous Series of Mediterranean Willow (*Ulmo canescentis-Salico pedicellatae* Sigmetum)

The riparian forest dominated by *Salix pedicellata* should be referred to the *Ulmo canescentis-Salicetum pedicellatae* Brullo & Spampinato 1990 (Column 9 in Table 1). The canopy of this vegetation, which can reach a height of 10–15 m. is characterized by the occurrence of *Salix alba*, *Populus alba*, *P. nigra* and *Ulmus canescens*. The *Alno-Populetea* P. Fukarek & Fabijanić 1968 class is represented by many species, such as *Carex pendula*, *Equisetum telmateia*, *Hypericum hircinum* subsp. *majus* and *Arum italicum* subsp. *italicum*.

This association is the most mature stage of an edapho-hygrophilous series. The degradation stages are mainly represented by shrubby communities belonging to *Pyro spinosae–Rubetalia ulmifolii* Biondi, Blasi & Casavecchia in Biondi et al. 2014 (*Crataego-Prunetea* class), such as *Rubo ulmifolii–Tametum communis* R. Tx. in R. Tx. & Oberd. 1958 or *Roso sempervirentis–Rubetum ulmifolii* Blasi, Di Pietro & Fortini 2000.

This azonal vegetation has its optimal conditions at altitudes between 300 and 800 m, within the thermo and mesomediterranean belts. Its geographical distribution includes several rivers and streams of northern and central Sicily [67], including the Giardinello watercourse within the investigated area.

The analysis of the woody components of current plant communities considered 56 taxa among the phanerophytes and nanophanerophytes of the nine forest and preforest associations recognized in the area (Table 2).

The analysis of the frequency of taxa, based on presence–absence data, highlights the ubiquitous nature of many species that are not exclusive to one association. Among the dominant trees, *Quercus ilex* (with a frequency equal to 1.00 in Holm Oak woods) has a frequency value of 0.40 and 0.67 in Downy Oak woods (*Oleo oleaster-Quercetum virgilianae* and *Sorbo torminali-Quercetum virgilianae*).

**Table 2.** Frequency of taxa and mean values of Braun-Blanquet's cover data of relevés for each association (A-Qi = *Ampelodesmo mauritanici-Quercetum ilicis*; S-Qi = *Sorbo torminalis-Quercetum ilicis*; O-Qv = *Oleo oleaster-Quercetum virgilianae*; S-Qv = *Sorbo torminalis-Quercetum virgilianae*; E-Ps = *Euphorbio characiae-Prunetum spinosae*; R-Ps = *Roso siculae-Prunetum spinosae*; R-Ru = *Roso corymbiferae-Rubetum ulmifolii*; Cl = *Crataegetum laciniatae*; U-Sp = *Ulmo-Salicetum pedicellatae*). The values of diagnostic species of association and alliance are in the grey boxes.

| | | | Forest Vegetation | | | | | | | | Shrubby Vegetation | | | | | | | | Riparian veg. | |
|---|---|---|---|---|---|---|---|---|---|---|---|---|---|---|---|---|---|---|---|---|
| | | Taxa | A-Qi | | S-Qi | | O-Qv | | S-Qv | | E-Ps | | R-Ps | | R-Ru | | Cl | | Sa-p | |
| id | L. f. | | freq. | cov. | freq. | cov. | freq. | cov. | freq. | cov. | freq. | cov. | freq. | cov. | freq. | cov. | freq. | cov. | freq. | cov. |
| | | **Characteristics and differentials of association and subassociation *Ampelodesmo mauritanici-Quercetum ilicis*** | | | | | | | | | | | | | | | | | | |
| 1 | P SCAP | *Quercus ilex* L. | 1.00 | 77.58 | 1.00 | 73.66 | 0.40 | 0.62 | 0.67 | 4.85 | 0.14 | 0.62 | . | . | . | . | . | . | . | . |
| 2 | NP | *Hippocrepis emerus* (L.) Lassen subsp. *emeroides* (Boiss. & Spruner) Lassen | 0.78 | 1.47 | . | . | 0.80 | 16.41 | . | . | 0.14 | 0.74 | . | . | . | . | . | . | 0.33 | 0.49 |
| 3 | P LIAN | *Lonicera implexa* Aiton | 0.70 | 2.39 | 0.28 | 0.64 | . | . | . | . | . | . | . | . | . | . | . | . | . | . |
| 4 | P CAESP | *Viburnum tinus* L. | 0.59 | 19.43 | . | . | . | . | 0.11 | 0.74 | . | . | . | . | . | . | . | . | . | . |
| 5 | P CAESP | *Arbutus unedo* L. | 0.52 | 5.20 | 0.11 | 0.62 | . | . | . | . | . | . | . | . | . | . | . | . | . | . |
| | | **Characteristics and differentials of association and subassociation *Sorbo torminalis-Quercetum ilicis*** | | | | | | | | | | | | | | | | | | |
| 6 | P SCAP | *Acer campestre* L. | . | . | 1.00 | 5.09 | . | . | 0.33 | 0.66 | . | . | 0.13 | 0.74 | . | . | 0.43 | 33.98 | . | . |
| 7 | P CAESP | *Daphne laureola* L. | 0.04 | 0.74 | 0.33 | 0.62 | . | . | 0.78 | 4.26 | 0.14 | 0.62 | 1.00 | 14.35 | 0.50 | 6.84 | 0.71 | 11.58 | . | . |
| | | **Characteristics and differentials of association and subassociation *Oleo oleaster-Quercetum virgilianae*** | | | | | | | | | | | | | | | | | | |
| 8 | P SCAP | *Quercus virgiliana* (Ten.) Ten. | 0.70 | 6.20 | 0.39 | 6.60 | 1.00 | 73.72 | 1.00 | 70.79 | . | . | . | . | . | . | . | . | . | . |
| | | **Characteristics and differentials of association and subassociation *Sorbo torminalis-Quercetum virgilianae*** | | | | | | | | | | | | | | | | | | |
| 9 | P CAESP | *Sorbus torminalis* (L.) Crantz | . | . | 0.44 | 6.43 | . | . | 1.00 | 5.30 | . | . | . | . | . | . | . | . | . | . |
| | | **Characteristics of the alliance *Fraxino-Quercion ilicis* and of the upper units** | | | | | | | | | | | | | | | | | | |
| 10 | NP | *Asparagus acutifolius* L. | 1.00 | 1.88 | 0.61 | 1.40 | 0.60 | 0.66 | 0.67 | 0.66 | 0.79 | 6.44 | 0.50 | 0.55 | 0.88 | 7.35 | 0.29 | 0.62 | . | . |
| 11 | NP | *Smilax aspera* L. | 1.00 | 7.95 | 0.44 | 2.61 | 0.80 | 4.78 | 0.33 | 0.66 | 0.36 | 25.15 | . | . | 0.63 | 0.54 | 0.14 | 0.49 | . | . |
| 12 | P LIAN | *Rubia peregrina* L. subsp. *longifolia* O. Bolòs | 0.93 | 0.62 | 0.50 | 0.52 | 0.80 | 0.62 | 0.67 | 2.03 | 0.64 | 0.60 | . | . | 0.75 | 3.31 | . | . | 0.67 | 0.49 |
| 13 | P SCAP | *Fraxinus ornus* L. | 0.67 | 7.69 | 0.78 | 5.20 | 0.20 | 0.74 | 0.33 | 0.74 | . | . | 0.13 | 0.74 | . | . | . | . | . | . |
| 14 | NP | *Osyris alba* L. | 0.48 | 4.85 | 0.17 | 0.49 | 0.60 | 3.48 | . | . | 0.43 | 2.03 | . | . | . | . | . | . | 1.00 | 0.57 |
| 15 | P CAESP | *Pistacia terebinthus* L. | 0.41 | 1.44 | . | . | . | . | . | . | . | . | . | . | . | . | . | . | . | . |
| 16 | NP | *Rosa sempervirens* L. | 0.37 | 0.59 | 0.17 | 0.66 | 0.20 | 0.49 | 0.44 | 2.73 | 0.29 | 2.67 | . | . | 0.13 | 0.49 | . | . | . | . |
| 17 | P CAESP | *Pistacia lentiscus* L. | 0.26 | 0.67 | . | . | . | . | . | . | . | . | . | . | . | . | . | . | . | . |
| 18 | P LIAN | *Lonicera etrusca* Santi | 0.19 | 0.54 | 0.83 | 1.14 | 0.40 | 0.62 | 0.67 | 1.98 | 0.29 | 10.19 | 0.13 | 0.74 | 0.75 | 0.70 | 0.29 | 0.62 | . | . |
| 19 | NP | *Cistus creticus* L. subsp. *creticus* | 0.15 | 0.49 | 0.28 | 0.64 | 0.60 | 3.39 | 0.33 | 0.49 | . | . | . | . | . | . | . | . | . | . |
| 20 | P SCAP | *Chamaerops humilis* L. | 0.19 | 0.64 | . | . | . | . | . | . | . | . | . | . | . | . | . | . | . | . |
| 21 | P CAESP | *Daphne gnidium* L. | 0.04 | 0.74 | . | . | . | . | . | . | . | . | . | . | . | . | . | . | . | . |
| 22 | P CAESP | *Myrtus communis* L. | 0.04 | 0.74 | . | . | . | . | . | . | . | . | . | . | . | . | . | . | . | . |
| 23 | P CAESP | *Calicotome infesta* (C. Presl) Guss. subsp. *infesta.* | . | . | . | . | 0.20 | 0.74 | . | . | . | . | . | . | . | . | . | . | . | . |
| 24 | P SCAP | *Quercus amplifolia* Guss. | . | . | . | . | . | . | 0.11 | 0.74 | . | . | . | . | . | . | . | . | . | . |
| | | **Characteristics and differentials of association and subassociation *Euphorbio characiae-Prunetum spinosae* e *Roso siculae-Prunetum spinosae*** | | | | | | | | | | | | | | | | | | |
| 25 | NP | *Euphorbia characias* L. | 0.41 | 1.37 | 0.33 | 0.53 | . | . | . | . | 1.00 | 7.33 | . | . | . | . | . | . | . | . |
| 26 | P CAESP | *Prunus spinosa* L. | 0.19 | 0.59 | 0.33 | 0.62 | 0.20 | 0.49 | 0.33 | 0.66 | 0.71 | 71.84 | 1.00 | 74.19 | 0.88 | 3.09 | 0.57 | 4.78 | . | . |
| 27 | NP | *Rosa sicula* Tratt. | . | . | . | . | . | . | 0.22 | 0.62 | . | . | 1.00 | 9.60 | . | . | 0.71 | 3.92 | . | . |
| 28 | NP | *Rosa glutinosa* Sm. | . | . | . | . | . | . | . | . | . | . | 0.88 | 0.67 | . | . | 0.29 | 0.62 | . | . |

**Table 2.** *Cont.*

| | | | Forest Vegetation | | | | | | | | | | Shrubby Vegetation | | | | | | Riparian veg. | |
|---|---|---|---|---|---|---|---|---|---|---|---|---|---|---|---|---|---|---|---|---|
| | | Taxa | A-Qi | | S-Qi | | O-Qv | | S-Qv | | E-Ps | | R-Ps | | R-Ru | | Cl | | Sa-p | |
| | | | *freq.* | *cov.* | *freq.* | *cov.* | *freq.* | *cov.* | *freq.* | *cov.* | *freq.* | *cov.* | *freq.* | *cov.* | *freq.* | *cov.* | *freq.* | *cov.* | *freq.* | *cov.* |
| | | **Characteristics and differentials of association and subassociation** *Roso corymbiferae-Rubetum ulmifolii* | | | | | | | | | | | | | | | | | | |
| 29 | NP | *Rubus ulmifolius* Schott | 0.26 | 0.53 | 0.33 | 0.62 | 0.80 | 2.73 | 0.67 | 3.48 | 0.93 | 5.52 | 0.75 | 0.53 | 1.00 | 44.69 | 0.14 | 0.74 | 0.33 | 0.49 |
| 30 | P CAESP | *Crataegus monogyna* Jacq. var. *monogyna* | 0.63 | 1.11 | 0.44 | 0.58 | 0.20 | 0.49 | 0.56 | 4.02 | 0.79 | 1.44 | . | . | 1.00 | 31.32 | 0.43 | 6.13 | . | . |
| 31 | NP | *Rosa corymbifera* Borkh. | . | . | 0.06 | 0.49 | . | . | . | . | 0.50 | 3.09 | . | . | 1.00 | 0.62 | 0.43 | 0.74 | . | . |
| | | **Characteristics and differentials of association and subassociation** *Crataegetum laciniatae* | | | | | | | | | | | | | | | | | | |
| 32 | P CAESP | *Crataegus rhipidophylla* Gand. | . | . | . | . | . | . | . | . | . | . | 1.00 | 4.85 | 0.25 | 0.49 | 1.00 | 47.46 | . | . |
| | | **Characteristics and differentials of the alliance** *Pruno-Rubion ulmifolii* **and of the upper units** | | | | | | | | | | | | | | | | | | |
| 33 | NP | *Rosa canina* L. | 0.37 | 1.44 | 0.28 | 2.23 | 0.60 | 3.48 | 0.44 | 0.74 | 1.00 | 15.50 | 0.88 | 1.81 | 1.00 | 5.87 | 0.29 | 4.72 | . | . |
| 34 | P SCAP | *Pyrus spinosa* Forssk. | . | . | . | . | . | . | . | . | 0.57 | 0.71 | 0.13 | 0.49 | 0.88 | 4.26 | 0.71 | 14.91 | . | . |
| 35 | P LIAN | *Hedera helix* L. subsp. *helix* | 0.81 | 2.75 | 0.94 | 4.59 | 0.40 | 4.85 | 0.67 | 7.54 | 0.79 | 7.56 | 0.38 | 0.57 | 0.88 | 2.98 | 0.57 | 12.30 | 0.33 | 0.74 |
| 36 | NP | *Rosa micrantha* Sm. | . | . | . | . | . | . | . | . | . | . | . | . | 0.25 | 0.74 | . | . | . | . |
| 37 | NP | *Rosa balsamica* Besser | . | . | . | . | . | . | . | . | . | . | . | . | 0.13 | 0.49 | . | . | . | . |
| 38 | P LIAN | *Clematis vitalba* L. | 0.56 | 1.17 | 0.33 | 3.44 | 0.20 | 0.74 | 0.11 | 0.49 | 0.14 | 0.62 | 0.50 | 0.74 | . | . | 0.14 | 0.74 | . | . |
| 39 | P CAESP | *Rhamnus alaternus* L. | . | . | . | . | . | . | . | . | 0.07 | 0.74 | . | . | . | . | . | . | . | . |
| 40 | P CAESP | *Ulmus minor* Mill. | . | . | 0.06 | 8.95 | . | . | . | . | 0.07 | 0.74 | . | . | . | . | . | . | 0.67 | 0.74 |
| | | **Characteristics and differentials of the alliance** *Berberido aetnensis-Crataegion laciniatae* | | | | | | | | | | | | | | | | | | |
| 41 | NP | *Rubus canescens* DC. | . | . | 0.11 | 0.62 | . | . | 0.44 | 0.62 | . | . | 0.50 | 0.55 | . | . | 0.57 | 4.72 | . | . |
| | | **Characteristics and hygrophilous differentials of** *Salicetum albo-pedicellatae* **and the alliance** *Populion albae* **and upper** | | | | | | | | | | | | | | | | | | |
| 42 | P CAESP | *Salix pedicellata* Desf. | . | . | . | . | . | . | . | . | . | . | . | . | . | . | . | . | 1.00 | 49.98 |
| 43 | P SCAP | *Populus nigra* L. | . | . | . | . | . | . | . | . | . | . | . | . | . | . | . | . | 0.70 | 5.43 |
| 44 | NP | *Hypericum hircinum* subsp. *majus* N. Robson. | . | . | . | . | . | . | . | . | . | . | . | . | . | . | . | . | 0.33 | 0.49 |
| 45 | NP | *Solanum dulcamara* L. | . | . | . | . | . | . | . | . | . | . | . | . | . | . | . | . | . | 0.49 |
| | | **Other species** | | | | | | | | | | | | | | | | | | |
| 46 | P SCAP | *Sorbus domestica* L. | . | . | 0.28 | 9.89 | 0.20 | 8.95 | . | . | . | . | . | . | . | . | . | . | . | . |
| 47 | NP | *Erica multiflora* L. subsp. *multiflora* | 0.33 | 3.37 | 0.11 | 0.74 | . | . | . | . | . | . | . | . | . | . | . | . | . | . |
| 48 | P SCAP | *Malus sylvestris* Mill. | . | . | . | . | . | . | 0.22 | 0.74 | . | . | 0.13 | 0.74 | . | . | . | . | . | . |
| 49 | P CAESP | *Euonymus europaeus* L. | . | . | . | . | . | . | 0.22 | 0.74 | . | . | . | . | . | . | . | . | . | . |
| 50 | P CAESP | *Cytisus villosus* Pourr. | . | . | . | . | . | . | 0.11 | 0.74 | . | . | . | . | . | . | . | . | . | . |
| 51 | P CAESP | *Mespilus germanica* L. | . | . | . | . | . | . | 0.11 | 0.49 | . | . | . | . | . | . | . | . | . | . |
| 52 | P CAESP | *Sorbus graeca* (Lodd. ex Spach) Kotschy | . | . | . | . | . | . | 0.22 | 8.95 | . | . | . | . | . | . | . | . | . | . |
| 53 | P CAESP | *Rhus coriaria* L. | . | . | . | . | 0.20 | 0.74 | . | . | . | . | . | . | . | . | . | . | 0.33 | 0.74 |
| 54 | P CAESP | *Anagyris foetida* L. | . | . | . | . | . | . | . | . | 0.14 | 0.74 | . | . | . | . | . | . | . | . |
| 55 | P CAESP | *Crataegus laevigata* Jacq. | . | . | . | . | . | . | . | . | 0.14 | 0.74 | . | . | . | . | . | . | . | . |
| 56 | P CAESP | *Ostrya carpinifolia* Scop. | . | . | . | . | . | . | 0.33 | 26.09 | . | . | . | . | . | . | . | . | . | . |

In contrast, *Quercus virgiliana* (with a frequency equal to 1.00 in Downy Oak woods) has a frequency value of 0.70 in *Ampelodesmo-Quercetum ilicis* and a value of 0.39 in *Sorbo-Quercetum ilicis*. *Fraxinus ornus* is a species which is always present in the forest formations, except in the riparian forest, while *Sorbus torminalis* grows only in forests with a meso-supramediterranean bioclimate (Figure 4). *Acer campestre*, in the same bioclimatic belt, is quite common both in forest and in secondary shrublands, such as *Roso siculae-Prunetum spinosae* and *Crataegetum laciniatae*. In contrast, species such as *Smilax aspera*, *Prunus spinosa* and *Rosa canina* are present in all climatophilous associations.

The mean coverage of woody plants, phanerophytes and nanophanerophytes (Table 2 and Figure 5) is the information that allowed us to statistically determine the probability of finding (and therefore collecting) a species in a community. The mean coverage of *Quercus ilex* is 77.58% and 73.66% in *Ampelodesmo-Quercetum ilicis* and in *Sorbo-Quercetum ilicis*, respectively, and the value dramatically drops to 0.62% in *Oleo-Quercetum virgilianae* and 4.85% and in *Sorbo torminalis-Quercetum virgilianae*.

In contrast, Downy Oak (*Quercus virgiliana*) shows mean coverages of 73.72% and 70.79% in *Oleo-Quercetum virgilianae* and in *Sorbo-Quercetum virgilianae*, respectively, becoming less significant in *Ampelodesmo-Quercetum ilicis* (6.20%) and in *Sorbo torminalis-ilicis* (6.60%).

In the forest associations, in addition to the oaks, which are the most abundant species, *Hippocrepis emerus* subsp. *emeroides* (16.41% in *Oleo-Quercetum virgilianae*) and *Viburnum tinus* (19.43% in *Ampelodesmo-Quercetum ilicis*) show higher values of coverage. In the secondary shrubland, species such as *Crataegetum laciniatae*, *Crataegus rhipidophylla* (47.46%), *Acer campestre* (33.98%), *Pyrus spinosa* (3.92%) and *Hedera helix* subsp. *helix* (12.30%) are dominant.

More generally, thorny Rosaceae are the species that shape the physiognomy of secondary shrub communities. *Rubus ulmifolius* (44.69%) and *Crataegus monogyna* var. *monogyna* (31.32%) are abundant in *Roso corymbiferae-Rubetum ulmifolii*; *Prunus spinosa* is common in *Roso siculae-Prunetum spinosae* and *Euphorbio characiae-Prunetum spinosae* (with mean cover values equal to 74.19% and 71.84%, respectively). In this latter association, *Rosa canina* shows a significant mean cover equal to 15.50%.

In the riparian forest dominated by *Salix pedicellata* (mean cover = 49.98%), *Populus nigra* is the second most abundant species in the association.

**Table 3.** Frequency of taxa per chronological phase: Phase 1, late 8th–9th century AD; Phase 2, 10th–11th century AD. Each column shows the indication of the absolute number of wood charcoals, their volumes in mm$^3$ and their incidence as a percentage of the total per phase.

| Taxa | | Phase 1 | | | Phase 2 | | |
|---|---|---|---|---|---|---|---|
| | | *Count* | *Vol (mm³)* | *Freq.* | *Count* | *Vol (mm³)* | *Freq.* |
| *Fagaceae* | *Quercus ilex* | 97 | 6350 | 0.55 | 23 | 1547 | 0.62 |
| *Fabaceae* | - | 1 | 30 | 0.01 | 2 | 181 | 0.07 |
| *Aceraceae* | *Acer campestre* | . | . | . | 2 | 95 | 0.04 |
| *Anacardiaceae* | *Pistacia terebinthus* | 24 | 1161 | 0.10 | 4 | 40 | 0.02 |
| *Oleaceae* | *Fraxinus ornus* | 4 | 98 | 0.01 | . | . | . |
| *Fagaceae* | *Quercus virgiliana* | 11 | 647 | 0.06 | . | . | . |
| *Rosaceae* | *Sorbus torminalis* | 9 | 342 | 0.03 | . | . | . |
| *Corylaceae* | *Ostrya carpinifolia* | 6 | 447 | 0.04 | 3 | 120 | 0.05 |
| *Rosaceae* | *Prunus spinosa* | 2 | 64 | 0.01 | 2 | 42 | 0.02 |
| *Rosaceae* | - | 13 | 600 | 0.05 | 6 | 252 | 0.10 |
| *Rhamnaceae* | *Rhamnus alaternus* | 9 | 546 | 0.05 | 2 | 42 | 0.02 |
| *Ulmaceae* | *Ulmus minor* | 10 | 616 | 0.05 | 3 | 148 | 0.06 |
| *Salicaceae* | *Populus nigra* | 8 | 684 | 0.06 | 3 | 18 | 0.01 |

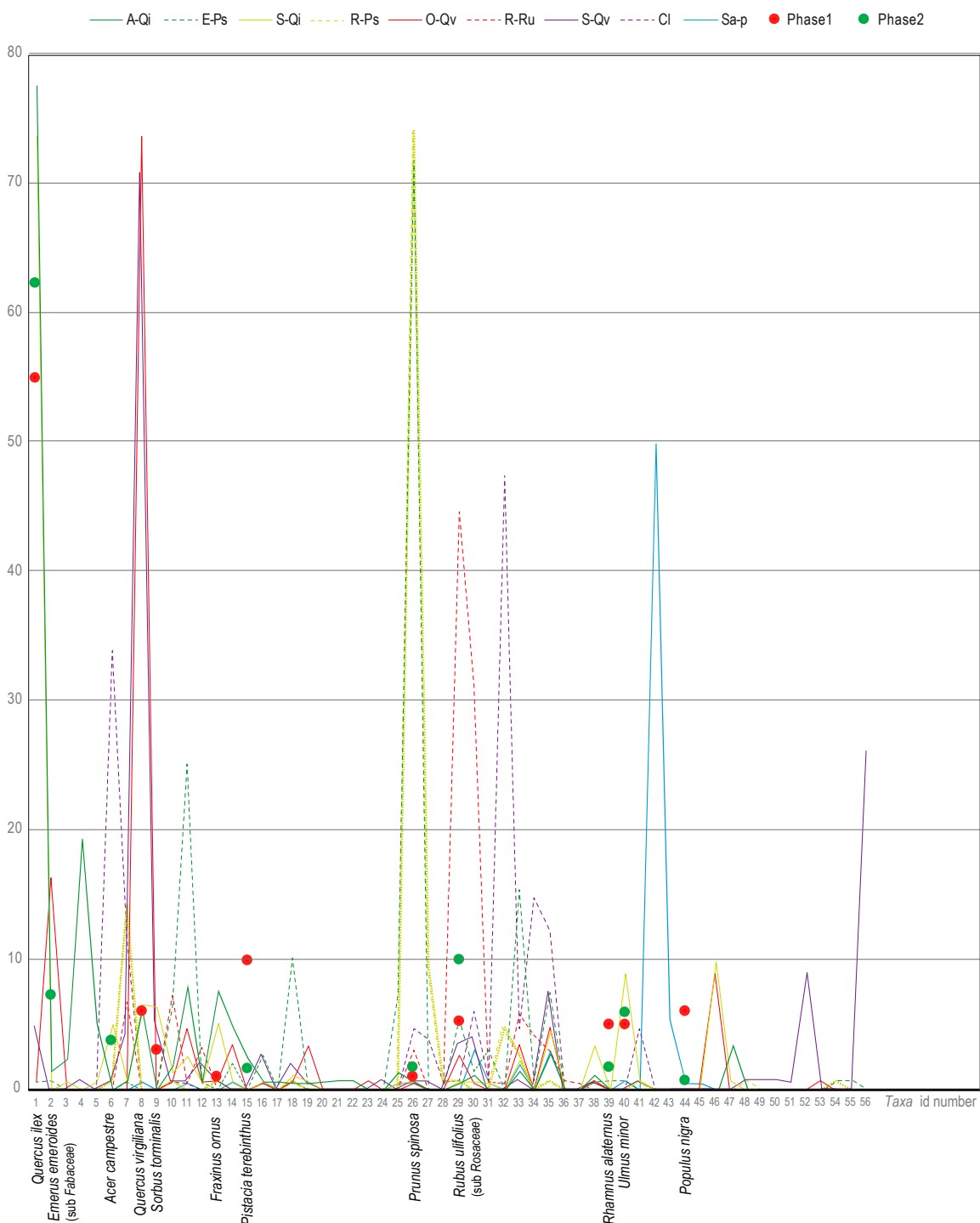

**Figure 5.** Mean coverage curves of the associations of Sicani Mountain forests and shrubs. The same color indicates associations belonging to the same vegetation series. The points show the incidence of identified taxa in the total volume of medieval wood charcoals (Phase 1: late 8th–9th century AD; Phase 2: 10th–11th century AD). In the x axis, the numbers indicate the taxa of Table 2 (phytosociological record) and the names are related to the taxa of Table 3 (archaeobotanical record).

### 4.2. Medieval Landscape: Anthracology and Wood Species

The preservation of the samples is quite good. Wood charcoals are from 1 mm$^3$ to about 1.5 cm$^3$ but on average quite small (3–4 mm$^3$). The volume of each stratigraphic unit in relation to the density of identified wood charcoals is quite similar (therefore the density is on average the same for all soils),

with the exception of a burnt layer connected to the fuel of the kilns (US 43) in which the density is strongly higher, some stratigraphic units (US 12, 13, 49, 52 and 61) in which it is higher and other stratigraphic units in which the density is clearly lower (US 36, 37, 1 and 2).

From the comparison with modern vegetation, nine species were identified (Quercus ilex, Quercus cfr. pubescens Willd., Pistacia terebinthus, Rhamnus alaternus, Fraxinus ornus, Ulmus minor, Acer campestre, Ostrya carpinifolia, Populus nigra); the identification rate reached the detail of genus or subfamily in four cases (Phillyrea sp., Sorbus sp., Prunus sp., cfr. Rosaceae, Pyrus sp., cfr. Fabaceae Hippocrepis emerus subsp. emeroides/Anagyris foetida). The arboreal vegetation is therefore represented by evergreen oaks, semi and deciduous oaks, maples, ash, and it is associated with riparian species such as elm, poplar and hornbeam and with shrubby species such as alatern, terebinth, rowan, plum.

In the phase of the late 8th–9th centuries AD, Holm Oak prevailed with 35% of the samples and, together with the other oaks, it made up 48% of the total. The terebinth followed with 9% and all other species exhibited between 1% and 5% of representativeness. In the furnace, many wood charcoals were mixed with ash concentrations at various densities; some of these fragments bore traces of strong distortion due to combustion processes. It is also interesting that, in this context, most of the species represented were Holm Oak and terebinth, with a very low percentage of other species.

In the second medieval phase (10th–11th centuries AD), the variability seemed to be slightly wider with 11 species/genera but with a clear preponderance in the representativeness of Holm Oak, which in terms of volume represented 49% of the total, followed by lower percentages of unidentified oaks, terebinth, poplar, elm and a leguminous plant perhaps identifiable as *Anagyris foetida*.

The relative frequencies were calculated by excluding the data on the unidentified samples and on those oaks that were not better identified (that is, they were not recognized either as deciduous or evergreen) and which therefore could not be placed more precisely within the series. The frequencies are quite similar in both phases, with a slight deviation in the incidence of Holm Oak (which increased in phase 2) and the absence of deciduous oaks in the more recent phase (Table 3).

The frequencies in the anthracological record were overlapped to the mean coverage of taxa (Figure 5). This overlapping has shown the correspondence between the species recorded in archaeobotanical samples with forest or preforest associations detected in the current area.

## 5. Discussion

This work compares the vegetation pattern of two anthropized landscapes in two specific temporal frames: the medieval period and the present day. The archaeobotanical data are an important source for understanding the effective use of wood resources in a specific place (Contrada Castro site) and time (Middle ages). However, these anthracological data need to be contextualized in a landscape model. The phytosociological approach has allowed us to model the plant communities as associations (depicted as tables) for creating a comparative reference for the archaeobotanical data.

In this way, the occurrence of species in the archaeobotanical record—in terms of charcoal volumes (Table 3)—could be related to the mean coverage of woody species in the association (Table 2) which indicated the probability of finding the species. Then, each association has been assigned to a specific vegetation series that is geographically spatialized in the current landscape.

The intersection between the frequency data of the archaeobotanical record and the phytosociological analysis of current vegetation—in terms of vegetation series—have firstly confirmed the maintenance of the same plant communities over the last millennium. In fact, the identified species in the anthracological sample have shown a coherent fitting with the data of current vegetation. Wood charcoal assemblage is not "anomalous", because all the species identified are present today in the case study area. Furthermore, the correlation of the archaeobotanical data with the plant associations and the related series (Figure 5) have provided ecological and environmental information.

This consistency between the occurrences in the archaeobotanical record and the vegetational pattern is linked to the fact that, over the last millennium, no radical changes occurred, and so

the ecological patterns of the vegetation series have not changed. In fact, this ecological pattern is determined by environmental or abiotic factors (e.g., climate, lithology, landforms) that change only over a long period of time [68] which is measurable on the scale of geological eras.

This long continuity in the vegetation pattern is also highlighted by the presence, as dominant species in these forest formations, of oaks, which are secular trees as regards their own physiology, as testified by the widespread presence in the Sicani mountains [69] of numerous enormous individuals with biological cycles of 400–500 years.

The archaeobotanical record has indicated an exclusive use of wood (as building materials or fuel) from the spontaneous plants specific to the natural vegetation of this area; no wood charcoals undoubtedly connected to cultivated tree plants have been found at the current stage of the investigation. Furthermore, according to the preliminary study of the seeds, the presence of fruits from fruit trees was not identified, apart from a single grape seed [70].

Previous analysis carried out on the catchment area of the Contrada Castro site by Bazan et al. [71] showed a 90% correlation with the distribution area of the Downy Oak series, which in Sicily is agriculturally suitable for arable land. The remaining 10% of the total catchment area of the site falls on the surface covered by the Holm Oak series. Holm Oak, according to the archaeobotanical record, is the most exploited species with percentages ranging between 30% and 50%. The massive use of wood species from different stages of the Holm Oak series clearly indicates an exploitation of the range of this series for forestry.

The high frequency of Rosaceae, which are typical of the secondary aspects of the forest (Figure 5, dotted line), is an indicator both of human activities of wood cutting and the use of this spiny rosacea species for combustible material for kilns and hearths, as identified in the archaeological excavation.

The use of the Monte Barraù area for wood exploitation for forestry, at least during the medieval period, is also attested to by a Latin parchment from AD 1428 from the *Tabularium* of the monastery of Santa Maria del Bosco di Calatamauro, which has owned the area of Barraù at least since the Late Middle Ages [47]. Preliminary data on the zooarchaeological sample from the excavation in Contrada Castro [72] have revealed, within a greater attestation of domestic species, a sporadic presence of wild taxa such as deer (*Cervus elaphus* L. 1758) and wild boar (*Sus scrofa* L. 1758). In particular, deer are no longer present in Sicily and appear to have become extinct between the 17th and 19th centuries [73]. By examining the ecological habits and habitat preferences of deer in Mediterranean areas where the species is still present, such as in Sardinia [74], the range of browsing activity of this animal during autumn, winter and spring is concentrated in the tall scrubwood area, and during the warm season, it moves to areas characterized by riparian vegetation, low scrubwood and oak woods. The presence of this animal is so strictly linked to its specific habitat that it allows us to hypothesize an extension and continuity of the wood to a relevant degree which is higher than currently.

The low percentage of wood elements attributable to the Downy Oak series indicates the scarce use of this type of wood resource, which is probably due to the fact that the Downy Oak area was intensely deforested in order to make way for arable lands. The agricultural exploitation was also documented by the discovery of charred seeds—still under study—which are similar, as can be seen from the medieval layers of Contrada Castro, to varieties of cereals and legumes connected precisely to the cultivation of the surrounding territory.

## 6. Conclusions

We believe that it is possible to reflect on the potentialities of this type of approach in the interpretation of current High Nature Value (HNV) farmlands.

In fact, the knowledge of the historical modes for the selection and exploitation of forest resources turns out to be a fundamental element in validating the characteristics and potentialities observable in the current landscape from geobotanical and phytosociological perspectives. It is necessary to return to the concept of the series of vegetation as an indicator of the dynamics of human–environment interactions. The passages between different stages of the succession of a series are, as a matter of

fact, a direct effect of anthropic activities (the cutting of the forest, fires, exploitation of pastures, agricultural activities). The comparison with the archaeobotanical data not only confirms various types of exploitation of wood resources but provides an insight into the past associations of plant communities and dynamics of the anthropic impact, which is observable even today. Basically, the window on the past opened by the archaeobotanical record allows us to envision a dynamic relationship between the suitability of this territory and the sustainability of its effective anthropic use into the last millennium. In this area of the Sicani Mountains, the anthropic exploitation has not altered the composition of the plant communities, maintaining a high degree of biodiversity, which favors the conservation of this landscape and therefore its designation as HNV farmland. For the designation of a certain landscape within the HNV farmland category, we believe it is necessary to evaluate its historical value, which is only defined by an integrated approach of historical ecology and environmental archaeology and can help us to identify the maintenance of geobotanical characteristics, biodiversity and the sustainability of exploitation of environmental resources over the long term.

**Supplementary Materials:** The following are available online at http://www.mdpi.com/2071-1050/12/8/3201/s1, Table S1: Phytosociological tables of associations of Sicani Mountains used as a data source.

**Author Contributions:** Conceptualization, G.B., A.C.B.; methodology, G.B., C.S.; archaeological investigation, A.C.B., R.M.; archaeobotanical investigation, C.S.; phytosociological investigation, G.B., P.M., S.C.; GIS mapping, G.B; writing—review and editing, G.B., A.C.B., C.S., S.C.; project administration, G.B., P.M.; funding acquisition, P.M. All authors have read and agreed to the published version of the manuscript.

**Funding:** This work was supported by the "Harvesting Memories: Ecology and Archaeology of Monti Sicani landscapes (Central-Western Sicily)" project, funded by Bona Furtuna LLC.

**Acknowledgments:** This study was supported by funds provided by Bona Furtuna LLC. The archaeological excavation has been carried out under the scientific direction of the Soprintendenza BB.CC.AA. of Palermo. A. C. B. thanks the Spanish MINECO for a Juan de la Cierva-Incorporación research fellowship (IJCI-2017-31494).

**Conflicts of Interest:** The authors declare no conflicts of interest.

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
