# Peer review of "Historical Suitability and Sustainability of Sicani Mountains Landscape (Western Sicily): An Integrated Approach of Phytosociology and Archaeobotany"

_sustainability, doi:10.3390/su12083201_

Round 1
Reviewer 1 Report
The paper presents an interesting comparison between nowadays and Medieval vegetation in a study area in Sicily. The manuscript is based on robust data set (106 relevés and 436 wood charcoal samples) and it may contribute to clarifies the role of the anthropic activities in the long-term landscape dynamics. So, the manuscript worth to be published. Nevertheless, the text is quite complex for researcher not skilled in phytosociological studies, especially because it is targeted at a wider community of researchers (e.g., landscape ecologist, anthropologist, climate change experts, etc.). I understand that it is hard to simplify a complex and multidisciplinary study, but to do this will strongly increase the interest of this job. For example, the paragraph "3.1 The current plant landscape" is quite long and complex for readers not skilled in phytosociology. Because your manuscript point at a wider community, please consider putting at the end of this paragraph a short summary of your main results that enable a direct comparison with the following part "3.2. Archaeobotany: Anthracology and Wood Species" and that are relevant for discussion. Otherwise, you might present past before the present in order to make easy to stress the similarity and to clearly explain the evolution of vegetation. Similarly, discussion may be improved by discussion deeper the similarity (and where appropriate the difference) between past and current landscape in the context of climatic and socio-economic changes. In fact, even if authors discuss this topic its discussion is a little too stylised, especially for readers that do not know the study area. Moreover, in discussion some methodological parts may be deleted. In this way, the readers can follow the main line of reasoning more easily.
Minor concerns
A map of Sicani Mountains in addition to the map of their position in Sicily may make more easy to understand some aspect of your work. Moreover, in the text you cite some localities in the Sicani Mountains that are probably unknown to the majority of reader. Put these localities in the geographical context would be useful
L72-82. This is a very key point of your paper. Vegetation enables us to interpret the spatial patterns and the temporal dynamics of landscape intended as the phytocoenosis that have been occurred over time in relation to anthropic activities. However, these analyses are just limited to a time-range of just over a century. Your strangeness is to analyse a long-time series. However, temporal dynamics (the series of phytocoenosis) may not be due to human activities only, particularly in long-time series, as you are aimed to study. In these series, natural factors like climatic change may have played a role interacting with anthropic activities. Because this point is very central in your line of argument you have to explain more clearly this part and the factors driving temporal dynamics. Are you able to discern between natural and anthropic factors?
L19-20. Please consider changing with “the ongoing project “Harvesting Memories” has been focused on long-term landscape dynamics in Sicani Mountains (Western Sicily)”
L23. Please consider changing “through” with “analysing”
L32. Please delete (HNV). The use of this abbreviation is not useful in the abstract where it is never used. Moreover, it is reported at the beginning of introduction and it was reported throughout the text together with its full-length form. You may consider using only abbreviation (after a first time you used it together with its full-length form) or full-length form.
L57. Please consider changing “faithful” with “useful”. Moreover, this sentence sounds a little odd. Vegetation is the description of a plant community and it is clear that may describe a landscape or that its changes may describe landscape transformation. However, it is not clear how it can be useful in the interpretation (explain causal factors) of this pattern and/or transformation. For example, a change from pasturage to forest may be due both to a grazing abandonment and to a forest upward shift because of climate change. As you correctly said at lines 61-62. So, is it this sentence truly useful? Can it be changed in a less compromising way, like: Vegetation studies are useful to describe landscape patterns and transformations. This seems to be more in line with your line of argument. Vegetation may describe landscape and its change. Nevertheless, it may be affected both by natural and human factors.
L287 double comma after Smilax aspera
L323 space between th and is in this
L360-365 Usually it is more advisable to avoid one-sentence paragraphs. Moreover, the sentence at L362-363 is not a true result. If useful, you might move it in M&M. Eventually, the sentence at L364-365 is not very useful in the current shape. What is the information in table 2 and fig 5 relevant for your discussion? Put a short summary of relevant results.
L386-390. This sentence is quite hard to understand. Please consider rephrasing it, possibly using brackets. E.g., “The volume of each stratigraphic unit in relation to the density of identified wood charcoals is quite similar (therefore the density is on average the same for all soils), with the exception of a burnt layer connected to the fuel of the kilns (SU 43) in which the density is strongly higher, some stratigraphic units (SU 12, 13, 49, 52 and 61) in which it is higher and other stratigraphic units in which the density is clearly lower (SU 36, 37, 1 and 2)”
L431-435. This paragraph is a description of you approach that was already previously described and does not seems very useful for the discussion. I suggest you to delete it to make discussion simpler and more focused on the results.
L464. The correlation analysis lacks in M&M (which method: parametric, non-parametric?) and in Results. In addition, all the paragraph (L463-469) seems a presentation of results
L475 has instead of have? And delete one of the two since in “since at least since”
Author Response
The paper presents an interesting comparison between nowadays and Medieval vegetation in a study area in Sicily. The manuscript is based on robust data set (106 relevés and 436 wood charcoal samples) and it may contribute to clarifies the role of the anthropic activities in the long-term landscape dynamics. So, the manuscript worth to be published.
Ans: thank you for your comments, we have accepted all your suggestions. The corrections heve been highlighted with blue text in the manuscript.
Nevertheless, the text is quite complex for researcher not skilled in phytosociological studies, especially because it is targeted at a wider community of researchers (e.g., landscape ecologist, anthropologist, climate change experts, etc.). I understand that it is hard to simplify a complex and multidisciplinary study, but to do this will strongly increase the interest of this job. For example, the paragraph "3.1 The current plant landscape" is quite long and complex for readers not skilled in phytosociology. Because your manuscript point at a wider community, please consider putting at the end of this paragraph a short summary of your main results that enable a direct comparison with the following part "3.2. Archaeobotany: Anthracology and Wood Species" and that are relevant for discussion.
Ans: In the paragraph 3.1 has been added a short summary (from line 241 to 252). The headings inside the description of the series have been removed in order to made more readable the text.
Otherwise, you might present past before the present in order to make easy to stress the similarity and to clearly explain the evolution of vegetation. Similarly, discussion may be improved by discussion deeper the similarity (and where appropriate the difference) between past and current landscape in the context of climatic and socio-economic changes. In fact, even if authors discuss this topic its discussion is a little too stylised, especially for readers that do not know the study area. Moreover, in discussion some methodological parts may be deleted. In this way, the readers can follow the main line of reasoning more easily.
Ans: This work aims to present the importance of correlation between current landscape and past landscape. So, as you suggested we changed the title of the paragraph 3.2 but we decided to maintain the order because the interpretation of archaeobotanical data is based on the current knowledge of vegetation.
Minor concerns
A map of Sicani Mountains in addition to the map of their position in Sicily may make more easy to understand some aspect of your work. Moreover, in the text you cite some localities in the Sicani Mountains that are probably unknown to the majority of reader. Put these localities in the geographical context would be useful
Ans: We have improved and modified the Figure 1
L72-82. This is a very key point of your paper. Vegetation enables us to interpret the spatial patterns and the temporal dynamics of landscape intended as the phytocoenosis that have been occurred over time in relation to anthropic activities. However, these analyses are just limited to a time-range of just over a century. Your strangeness is to analyse a long-time series. However, temporal dynamics (the series of phytocoenosis) may not be due to human activities only, particularly in long-time series, as you are aimed to study. In these series, natural factors like climatic change may have played a role interacting with anthropic activities. Because this point is very central in your line of argument you have to explain more clearly this part and the factors driving temporal dynamics. Are you able to discern between natural and anthropic factors?
Ans: From line 74 to 78 we have clarified this concept that was also discussed from line 470 to 478.
L19-20. Please consider changing with “the ongoing project “Harvesting Memories” has been focused on long-term landscape dynamics in Sicani Mountains (Western Sicily)”
Ans: changed.
L23. Please consider changing “through” with “analysing”
Ans: changed.
L32. Please delete (HNV). The use of this abbreviation is not useful in the abstract where it is never used. Moreover, it is reported at the beginning of introduction and it was reported throughout the text together with its full-length form. You may consider using only abbreviation (after a first time you used it together with its full-length form) or full-length form.
Ans: According to other referee indications, we decided to use only the abbreviation after the first mention.
L57. Please consider changing “faithful” with “useful”. Moreover, this sentence sounds a little odd. Vegetation is the description of a plant community and it is clear that may describe a landscape or that its changes may describe landscape transformation. However, it is not clear how it can be useful in the interpretation (explain causal factors) of this pattern and/or transformation. For example, a change from pasturage to forest may be due both to a grazing abandonment and to a forest upward shift because of climate change. As you correctly said at lines 61-62. So, is it this sentence truly useful? Can it be changed in a less compromising way, like: Vegetation studies are useful to describe landscape patterns and transformations. This seems to be more in line with your line of argument. Vegetation may describe landscape and its change. Nevertheless, it may be affected both by natural and human factors.
Ans: We accepted to insert this sentence “Vegetation studies are useful to describe landscape patterns and transformations”.
L287 double comma after Smilax aspera
Ans: corrected.
L323 space between th and is in this
Ans: corrected.
L360-365 Usually it is more advisable to avoid one-sentence paragraphs. Moreover, the sentence at L362-363 is not a true result. If useful, you might move it in M&M. Eventually, the sentence at L364-365 is not very useful in the current shape. What is the information in table 2 and fig 5 relevant for your discussion? Put a short summary of relevant results.
Ans: corrected.
L386-390. This sentence is quite hard to understand. Please consider rephrasing it, possibly using brackets. E.g., “The volume of each stratigraphic unit in relation to the density of identified wood charcoals is quite similar (therefore the density is on average the same for all soils), with the exception of a burnt layer connected to the fuel of the kilns (SU 43) in which the density is strongly higher, some stratigraphic units (SU 12, 13, 49, 52 and 61) in which it is higher and other stratigraphic units in which the density is clearly lower (SU 36, 37, 1 and 2)”
Ans: corrected.
L431-435. This paragraph is a description of you approach that was already previously described and does not seems very useful for the discussion. I suggest you to delete it to make discussion simpler and more focused on the results.
Ans: We move this sentence from paragraph 4 to paragraph 3.2 (lines 438-438).
L464. The correlation analysis lacks in M&M (which method: parametric, non-parametric?) and in Results. In addition, all the paragraph (L463-469) seems a presentation of results
Ans: This is an already published data, we insert the reference in the text in a more clear way (line 484-485).
L475 has instead of have? And delete one of the two since in “since at least since”
Ans: corrected.

Reviewer 2 Report
The manuscript presents an interesting view on the possibilities of human activity reconstruction based on current dynamics of plant communities and archaeobotanical record. The manuscript is well-written and clearly defined the research problem. However, the manuscript requires some additions:
- The archaeological site is very interesting and should be presented in more details from the physiography point of view for a wide audience, as the Sustainability is an international journal. The additional paragraph (in part 2. Archaeological background of the site, within 118 – 163 lines) should include e.g.: relief characteristics, soil conditions, exposure, altitude.
- The photographs are informative, but it is necessary to create new figure – morphological sketch showing the detailed location of the site (in part 2).
- sub-chapter 2.2.: additional information about size of charcoal is necessary
- sub-chapter 2.2.: what does it mean “stratigraphic unit” in this paper? Stratigraphic unit of soil? If yes, this is an incorrect statement. This term is usually used to describe the geological position of sediments and not soil levels/soil sample. Please rewrite it in all manuscript.
- The authors mentioned about radiocarbon analysis and refer to these dating in results. Did the charcoals are dated? What is the relationship between coals and settlement phases? The criticism I would make is that authors did not present clearly this issue. The authors combine charcoals with settlement phases but do not explain what material was dated. What is the proof for this age of charcoals? Please explain this in section 2.2 and in the results 3.2. Please refer to this issue also in the discussion.
Author Response
The manuscript presents an interesting view on the possibilities of human activity reconstruction based on current dynamics of plant communities and archaeobotanical record. The manuscript is well-written and clearly defined the research problem. However, the manuscript requires some additions:
Ans: thank you for your comments, we have accepted all your suggestions. The corrections have been highlighted with green text in the manuscript. An English proof-reading of the text has been provided by MDPI service.
The archaeological site is very interesting and should be presented in more details from the physiography point of view for a wide audience, as the Sustainability is an international journal. The additional paragraph (in part 2. Archaeological background of the site, within 118 – 163 lines) should include e.g.: relief characteristics, soil conditions, exposure, altitude.
Ans: We added this information from line 141 to 146.
The photographs are informative, but it is necessary to create new figure – morphological sketch showing the detailed location of the site (in part 2).
Ans: We have improved and modified the Figure 1.
sub-chapter 2.2.: additional information about size of charcoal is necessary
Ans: We inserted this information in lines 408-409.
sub-chapter 2.2.: what does it mean “stratigraphic unit” in this paper? Stratigraphic unit of soil? If yes, this is an incorrect statement. This term is usually used to describe the geological position of sediments and not soil levels/soil sample. Please rewrite it in all manuscript.
Ans: We have clarified this concept (archaeological layer) also adding a reference at line 204.
The authors mentioned about radiocarbon analysis and refer to these dating in results. Did the charcoals are dated? What is the relationship between coals and settlement phases? The criticism I would make is that authors did not present clearly this issue. The authors combine charcoals with settlement phases but do not explain what material was dated. What is the proof for this age of charcoals? Please explain this in section 2.2 and in the results 3.2. Please refer to this issue also in the discussion.
Ans: We have specified the chronology of radiocarbon dating in paragraph 2 and with a mention also in paragraph 3.2

Reviewer 3 Report
Thank you for letting me read and comment on this paper. My comments are highlighted and inserted in the attached pdf.
My general impression of the paper is that it is of interest, but the presentation of the data and the results must be improved. The language has to a large extent a very oral form (especially in the introduction and methods parts). Some sentences are very long, and even have a lot of inserted sentences. I have marked some. This should be looked into very carefully.
Focus should be set on presenting the important data in relation to the aims of the study.

Author Response
Thank you for letting me read and comment on this paper. My comments are highlighted and inserted in the attached pdf. My general impression of the paper is that it is of interest, but the presentation of the data and the results must be improved. The language has to a large extent a very oral form (especially in the introduction and methods parts). Some sentences are very long, and even have a lot of inserted sentences. I have marked some. This should be looked into very carefully. Focus should be set on presenting the important data in relation to the aims of the study.
Ans: thank you for your comments, we have accepted all your suggestions. The corrections have been highlighted with red text in the manuscript. An English proof-reading of the text has been provided by MDPI service.

Round 2
Reviewer 2 Report
Authors improved and clarified the manuscript. I accept it for publication.
Reviewer 3 Report
I think the paper is very much improved now. thank you for following the suggestions from all reviewers. It is much easier to read and to follow the red line in the paper.